# Bayesian Optimization from Human Feedback: Near-Optimal Regret Bounds

**Aya Kayal** [1]  **Sattar Vakili** [2]  **Laura Toni** [1]  **Da-shan Shiu** [2]  **Alberto Bernacchia** [2]

## Abstract

Bayesian optimization (BO) with preference-based feedback has recently garnered significant attention due to its emerging applications. We refer to this problem as Bayesian Optimization from Human Feedback (BOHF), which differs from conventional BO by learning the best actions from a reduced feedback model, where only the preference between two actions is revealed to the learner at each time step. The objective is to identify the best action using a limited number of preference queries, typically obtained through costly human feedback. Existing work, which adopts the Bradley-Terry-Luce (BTL) feedback model, provides regret bounds for the performance of several algorithms. In this work, within the same framework we develop tighter performance guarantees. Specifically, we derive regret bounds of $\tilde{\mathcal{O}}(\sqrt{\Gamma(T)T})$, where $\Gamma(T)$ represents the maximum information gain—a kernel-specific complexity term—and $T$ is the number of queries. Our results significantly improve upon existing bounds. Notably, for common kernels, we show that the order-optimal sample complexities of conventional BO—achieved with richer feedback models—are recovered. In other words, the same number of preferential samples as scalar-valued samples is sufficient to find a nearly optimal solution.

## 1. Introduction

Optimizing a black-box function using only preference-based feedback between pairs of candidate solutions has recently emerged as an interesting problem. This approach finds application, for instance, in prompt optimization (Lin

Aya Kayal's work was part of her research placement at MediaTek Research. [1]University College London, UK [2]MediaTek Research. Correspondence to: Aya Kayal <aya.kayal.21@ucl.ac.uk>, Sattar Vakili <sattar.vakili@mtkresearch.com>.

*Proceedings of the $42^{nd}$ International Conference on Machine Learning*, Vancouver, Canada. PMLR 267, 2025. Copyright 2025 by the author(s).

et al., 2024), which aims to efficiently identify the best prompt for black-box Large Language Models (LLMs), thereby significantly enhancing their performance (Chen et al., 2024; Lin et al., 2024; 2023). Obtaining a numeric score to evaluate each prompt's performance is often unrealistic, but human users are generally much more reliable at providing preference feedback between pairs of prompts (Lin et al., 2024). Since human feedback is costly, it becomes essential to develop efficient methods that can sequentially select favorable pairs of actions while minimizing the number of feedback instances required.

The theoretical framework for learning from preference-based feedback (see, e.g., Pásztor et al., 2024; Xu et al., 2024) can be modeled as Bayesian Optimization from Human Feedback (BOHF). Similarly to conventional BO (Frazier, 2018; Shahriari et al., 2015; Srinivas et al., 2010), the learner leverages previously collected samples through kernel-based regression to learn an unknown black-box function. However, unlike conventional BO methods that rely on direct evaluations of the target function, this approach collects pairwise comparisons instead of direct evaluation feedback, adding further complexities to the problem.

In the BOHF framework, at each time step $t = 1, 2, \cdots, T$, the learner selects a pair of actions $(x_t, x'_t)$ and receives binary feedback $y_t \in \{0, 1\}$ representing the preference between the two actions. This binary feedback is modeled as a Bernoulli random variable, where the parameter is determined by applying a link function (here, sigmoid) to the difference in the unobserved utilities corresponding to each action, quantifying the preference between them. Performance is measured in terms of regret, defined as the cumulative loss in the selected pairs of actions compared to the optimal action (details are provided in Section 2). Kernel-based models employed within the BOHF framework allow for powerful and versatile modeling of preferences among actions, leveraging structures, and handling continuous domains or very large action spaces.

Existing work establishes a regret bound of $\tilde{\mathcal{O}}\left(\Gamma(T)\kappa^2\sqrt{T}\right)$ for the BOHF problem (Pásztor et al., 2024), where we use the $\mathcal{O}$ and the $\tilde{\mathcal{O}}$ notations to hide constants and logarithmic terms respectively, for simplicity of presentation. In this expression, $\kappa$ is the maximum of the derivative of the inverse link function (see

Equation (2)) and $\Gamma(T)$ is the maximum information gain, a kernel-specific and algorithm-independent complexity term (see Equation (13)).

It is insightful to compare the existing BOHF regret bound with the order-optimal regret bounds of $\tilde{\mathcal{O}}\left(\sqrt{\Gamma(T)T}\right)$ in conventional BO. In comparison, an additional $\kappa^2$ factor arises due to the feedback model. While this constant is independent of $T$, it can be very large. There is also an extra $\sqrt{\Gamma(T)}$ factor, which introduces potential challenges. To better understand this, let us take a closer look at $\Gamma(T)$. For smooth kernels with exponentially decaying eigenvalues, such as the Squared Exponential (SE) kernel, $\Gamma(T)$ is polylogarithmic in $T$. However, for more general kernels of both practical and theoretical interest, such as the Matérn family (Borovitskiy et al., 2020) and Neural Tangent (NT) kernels (Arora et al., 2019), $\Gamma(T)$ grows polynomially with $T$, possibly faster than $\sqrt{T}$, making the regret bounds vacuous (linear in $T$).

Our contribution is that we establish regret bounds of $\tilde{\mathcal{O}}\left(\sqrt{\Gamma(T)T}\right)$ for the BOHF problem (Theorem 4.1), achieving a $\sqrt{\Gamma(T)}$ improvement and eliminating the dependency on $\kappa$, resolving both issues and matching the regret bounds of conventional BO. From our regret bounds, we derive the sample complexities—the number of preference query samples required to identify near-optimal actions. Our sample complexities match the lower bounds obtained in Scarlett et al. (2017) for conventional BO, which benefits from a richer feedback model with a different noise distribution. We will provide a technical discussion on this in Section 4.

In summary, we establish the intriguing result that the number of preferential feedback samples required to identify near-optimal actions is of the same order as the number of scalar-valued feedback samples. This is in sharp contrast and a significant improvement over the existing work (Pásztor et al., 2024; Xu et al., 2024).

To obtain the improved regret bounds, we propose an algorithm referred to as Multi-Round Learning from Preference-based Feedback (MR-LPF). The proposed algorithm proceeds in rounds. In each round, pairs of actions are sequentially selected based on the highest uncertainty in their preference. This method effectively reduces uncertainties about the preferences between actions by the end of each round. The uncertainties are represented by kernel-based standard deviations. At the end of each round, the kernel-based confidence intervals are used to eliminate actions unlikely to be the best. Our multi-round structure is inspired by the BPE algorithm of Li & Scarlett (2022), though the details and analysis differ significantly due to the preference-based feedback model. Details are provided in Section 3. We show that this structure allows for a more efficient use of kernel-

based confidence intervals, contributing to improvements in both $\Gamma(T)$ and $\kappa$.

We present experimental results on the performance of MR-LPF on synthetic functions that closely align with the analytical assumptions, as well as on a dataset of Yelp reviews, demonstrating the utility of the proposed algorithm in real-world applications (Section 5).

## 1.1. Related Work

Two works closely related to ours are Pásztor et al. (2024) and Xu et al. (2024), which consider the exact same BOHF framework. The work by Pásztor et al. (2024) proposed the MaxMinLCB algorithm, which takes a game-theoretic approach to selecting the pair of actions $(x_t, x_t')$ at each time step $t$. Specifically, $x_t$ and $x_t'$ are selected according to a game, with the objective function defined as a lower confidence bound (LCB) on the probability of favoring $x_t$ over $x_t'$. Hence, the name: $x_t$ is chosen to *Max*imize and $x_t'$ to *Min*imize the *LCB* (see, Pásztor et al., 2024, Algorithm 1). Their regret bound scales as $\tilde{\mathcal{O}}\left(\Gamma(T)\kappa^2\sqrt{T}\right)$, which may be vacuous for some commonly used kernels and scales with $\kappa^2$, which can be a large constant.

Another closely related work is Xu et al. (2024), which develops Principled Optimistic Preferential Bayesian Optimization (POP-BO), an algorithm based on the optimism principle. Specifically, at each time step $t$, $x_t'$ is set to $x_{t-1}$, one of the actions from the previous time step, and $x_t$ is set to the maximizer of an upper confidence bound on the preference between the two actions (see, Xu et al., 2024, Algorithm 1). They establish a regret bound of $\tilde{\mathcal{O}}\left((\Gamma(T)T)^{3/4}\right)$, which is larger than the one in Pásztor et al. (2024) by a factor of $(T/\Gamma(T))^{1/4}$ and similarly may be vacuous for many cases of interest.[1] Their definition of regret is based directly on the utility function and slightly differs from ours. However, it remains equivalent to our regret definition up to a constant factor, as discussed in Pásztor et al. (2024).

*Table 1.* Comparison of regret bounds in BOHF.

| (Pásztor et al., 2024) | (Xu et al., 2024) | **This work** |
|---|---|---|
| $\tilde{\mathcal{O}}\left(\Gamma(T)\kappa^2\sqrt{T}\right)$ | $\tilde{\mathcal{O}}\left((\Gamma(T)T)^{3/4}\right)$ | $\tilde{\mathcal{O}}\left(\sqrt{\Gamma(T)T}\right)$ |

Some other preferential BO methods mainly propose heuristics without formal theoretical guarantees on regret or convergence proofs (González et al., 2017; Mikkola et al., 2020; Takeno et al., 2023).

---

[1]Xu et al. (2024) does not explicitly report the scaling of the regret bound with $\kappa$.

### 1.1.1. CONVENTIONAL BO

Classical BO algorithms include strategies based on the upper confidence bound (UCB), Thompson sampling (TS) (Thompson, 1933), expected improvement (EI) (Jones et al., 1998), and probability of improvement (PI) (Brochu et al., 2010). A line of research has established regret bounds for BO algorithms, including the $\tilde{\mathcal{O}}(\Gamma(T)\sqrt{T})$ bounds for GP-UCB (Srinivas et al., 2010) and GP-TS (Chowdhury & Gopalan, 2017). Several works have achieved tighter $\tilde{\mathcal{O}}(\sqrt{\Gamma(T)T})$ bounds, including *Sup* variations of UCB (Valko et al., 2013), the domain-shrinking algorithm GP-ThreDS (Salgia et al., 2021), and Batch Pure Exploration (BPE) (Li & Scarlett, 2022). The latter also features a multi-round structure and has inspired our MR-LPF algorithm. However, there are differences in the inference procedure and analysis, due to the use of a reduced preference-based feedback model, which introduces additional complexities in both algorithm design and theoretical analysis.

### 1.1.2. DUELING BANDITS

The BOHF framework can be viewed as an extension of bandits with preference-based feedback, also known as dueling bandits (Yue & Joachims, 2009; Yue et al., 2012), where the goal is to identify the best action from a set of actions using only pairwise comparisons. For a comprehensive survey on dueling bandits, see Bengs et al. (2021). Dueling bandit problems focus on multi-armed settings and learning the pairwise preference matrix by applying noisy sorting or tournament algorithms (Ailon et al., 2014; Zoghi et al., 2014; Falahatgar et al., 2017; Zoghi et al., 2015). These approaches are typically limited to scenarios where the number of arms is small, and their regret can become unbounded as the number of arms approaches infinity. The simplest structured variation is the linear contextual dueling bandit, studied in Dudík et al. (2015); Saha & Krishnamurthy (2022); Das et al. (2024); Li et al. (2024); Bengs et al. (2022), which allows for a large number of actions but under the limiting assumption of linear structure. Several works have extended the dueling bandit problem to kernel-based settings, which differ from our BOHF framework. For instance, Xu et al. (2020a); Mehta et al. (2023a;b) consider the *Borda* score, representing the probability that a selected action is preferred over a uniformly sampled action from the domain. They make strong assumptions about the Borda function, which effectively reduces the problem to conventional BO. In contrast, our analytical requirements are significantly different from these approaches. A recent extension (Verma et al., 2025) considers neural dueling bandits with a wide neural network for preference prediction. Their approach differs in both modeling and action selection, with regret bounds depending on the model's effective dimension and the curvature parameter $\kappa$.

### 1.1.3. REINFORCEMENT LEARNING FROM HUMAN FEEDBACK (RLHF)

Another related line of work is RLHF (Griffith et al., 2013; Novoseller et al., 2020; Xu et al., 2020b; Wu & Sun, 2024; Saha et al., 2023; Chen et al., 2022), which has gained popularity due to its success in fine-tuning LLMs (Ouyang et al., 2022). In this context, preference-based feedback is provided for Markov decision process trajectories or policies rather than pairs of actions. However, these results are primarily limited to tabular (finite state-action) or linear settings and are not directly related to our kernel-based setting.

## 2. Preliminaries and Problem Formulation

In this section, we provide details of the BOHF framework. We also outline the methods used to predict preference functions and estimate uncertainty, which form the foundation of our algorithm's design and analysis.

### 2.1. BOHF Framework

At each step $t = 1, 2, \cdots, T$, the agent selects a pair of actions $x_t$ and $x'_t$, from the set $\mathcal{X}$, which can either be a continuous space or a (possibly very large) discrete set. We consider the following feedback model: Let $y_t \in \{0, 1\}$ be a binary random variable indicating the preference between $x_t$ and $x'_t$, defined as $y_t = \mathbb{1}\{x_t \succ x'_t\}$. The notation $x_t \succ x'_t$ denotes that action $x_t$ is preferred over action $x'_t$ and $\mathbb{1}$ is the indicator function. Specifically, following the existing work, for each pair $(x, x') \in \mathcal{X} \times \mathcal{X}$, the random variable $y = \mathbb{1}\{x \succ x'\}$ is modelled as a Bernoulli random variable satisfying $\mathbb{P}(y = 1|x, x') = \mu(f(x) - f(x'))$. Here, $\mu : \mathbb{R} \rightarrow [0, 1]$ is a known monotonically increasing link function satisfying $\mu(0) = \frac{1}{2}$ that is assumed to be the sigmoid function $\mu(\cdot) = (1 + e^{-\cdot})^{-1}$, and $f : \mathcal{X} \rightarrow \mathbb{R}$ is an unknown latent utility function that quantifies the value of each action. This preference feedback model is referred to as the Bradeley-Terry-Luce (BTL) model (Bradley & Terry, 1952) and is widely utilized in bandit and RL problems with preference feedback (Pásztor et al., 2024; Xu et al., 2024; Zhan et al., 2024; Wu & Sun, 2024).

We note that when $f(x) > f(x')$, we have $\mathbb{P}(x \succ x') = \mathbb{P}(y = 1|x, x') = \mu(f(x) - f(x')) > \frac{1}{2}$, and vice versa. We also emphasize that this feedback model is weaker than the standard BO where the per-step utility signal (the quantitative value of $f$) is revealed, typically as a scalar value.

The goal is to sequentially select favorable action pairs over a horizon of $T$ steps, and converge to the globally preferred action $x^\star$, defined as $x^\star = \arg\max_{x \in \mathcal{X}} f(x)$. A common objective adopted in the literature is to design an algorithm with sublinear cumulative regret over the horizon $T$, defined as the sum of the average sub-optimality gap between the

selected pair and the globally optimal action:

$$R(T) = \sum_{t=1}^{T} \frac{\mathbb{P}(x^\star \succ x_t) + \mathbb{P}(x^\star \succ x_t') - 1}{2}. \quad (1)$$

It can be shown that the value of regret above is equivalent to a variation of regret defined on the utility function: $\sum_{t=1}^{T} (f(x^\star) - (f(x_t) + f(x_t'))/2)$—used in Xu et al. (2024)—up to constants that depend on the link function (Saha, 2021).

The notion of regret accounts for the entire sequence of query points throughout steps $t = 1, 2, \ldots, T$. Alternatively, one may be interested solely in the final performance. In this case, the algorithm outputs $\hat{x}_T$ at the end of $T$ samples, and the performance is measured in terms of $\mathbb{P}(x^\star \succ \hat{x}_T) - \frac{1}{2}$. We refer to the number of samples $T$ required to ensure $\mathbb{P}(x^\star \succ \hat{x}_T) - \frac{1}{2} \leq \epsilon$, for some $0 < \epsilon < 1/2$, as the sample complexity and also remark on the sample complexity of different algorithms.

An important quantity that appears in the analysis is

$$\kappa = \sup_{x,x' \in \mathcal{X}} \frac{1}{\dot{\mu}(f(x) - f(x'))}, \quad (2)$$

where $\dot{\mu}$ denotes the derivative of the link function $\mu$ and $\kappa$ captures its curvature. The dependence on $\kappa$ has been extensively studied in linear logistic bandits, with recent works successfully removing the regret dependency on $\kappa$ (Faury et al., 2020). To emphasize the significance of this quantity, consider the case where $f$ is bounded within the interval $[-5, 5]$. In this scenario, $\kappa$ can become extremely large ($> 22028$). When the algorithm selects an action pair $(x, x')$ that are nearly equally favorable, $f(x) - f(x')$ will be close to 0, in which case the inverse derivative of the sigmoid function is almost a constant 4. However, when one action is clearly preferred over the other, $|f(x) - f(x')|$ becomes large, making the inverse derivative of the sigmoid function very large. Therefore, a crucial aspect of algorithm design is to remove the dependency on $\kappa$ defined in (2) by ensuring that the algorithm gradually queries only closely preferred actions.

### 2.2. Preliminaries and Assumptions

Similar to Pásztor et al. (2024); Xu et al. (2024), we assume that the utility function $f$ belongs to a known Reproducing Kernel Hilbert Space (RKHS). This is a very general assumption, considering that the RKHS of common kernels can approximate almost all continuous functions on the compact subsets of $\mathbb{R}^d$ (Srinivas et al., 2010) . Consider a positive definite kernel $k : \mathcal{X} \times \mathcal{X} \rightarrow \mathbb{R}$. Let $\mathcal{H}_k$ be the RKHS induced by $k$, where $\mathcal{H}_k$ contains a family of functions defined on $\mathcal{X}$. Let $\langle \cdot, \cdot \rangle_{\mathcal{H}_k} : \mathcal{H}_k \times \mathcal{H}_k \rightarrow \mathbb{R}$ and $\| \cdot \|_{\mathcal{H}_k} : \mathcal{H}_k \rightarrow \mathbb{R}$ denote the inner product and the norm

of $\mathcal{H}_k$, respectively. The reproducing property implies that for all $f \in \mathcal{H}_k$, and $x \in \mathcal{X}$, $\langle f, k(\cdot, x) \rangle_{\mathcal{H}_k} = f(x)$. Mercer theorem implies, under certain mild conditions, $k$ can be represented using an infinite dimensional feature map:

$$k(x, x') = \sum_{m=1}^{\infty} \gamma_m \varphi_m(x) \varphi_m(x'), \quad (3)$$

where $\gamma_m > 0$, and $\sqrt{\gamma_m} \varphi_m \in \mathcal{H}_k$ form an orthonormal basis of $\mathcal{H}_k$. In particular, any $f \in \mathcal{H}_k$ can be represented using this basis and weights $w_m \in \mathbb{R}$ as $f = \sum_{m=1}^{\infty} w_m \sqrt{\gamma_m} \varphi_m$, where $\|f\|_{\mathcal{H}_k}^2 = \sum_{m=1}^{\infty} w_m^2$. A formal statement and the details are provided in Appendix A. We refer to $\gamma_m$ and $\varphi_m$ as (Mercer) eigenvalues and eigenfeatures of $k$, respectively.

Let us use the notation $z = (x, x')$ and $h(z) = f(x) - f(x')$, for $(x, x') \in \mathcal{X} \times \mathcal{X}$. As shown in Pásztor et al. (2024), we can define a *dueling* kernel

$$\mathbb{k}(z_1, z_2) = k(x_1, x_2) + k(x_1', x_2') - k(x_1, x_2') - k(x_1', x_2), \quad (4)$$

where, we have: $\|f\|_{\mathcal{H}_k} = \|h\|_{\mathcal{H}_{\mathbb{k}}}$ (Pásztor et al., 2024, Proposition 4).

Below is a formal statement of our assumptions on $f$.

**Assumption 2.1.** We assume that the utility function is in the RKHS of a known kernel $k$ satisfying $\|f\|_{\mathcal{H}_k} \leq B$ for some constant $B > 0$. Without loss of generality, we assume that the kernel function is normalized $k(., .) \leq 1$ everywhere in the domain.

### 2.3. Preference Function Prediction and Uncertainty Estimation

The preference-based feedback model in BOHF is weaker than the standard BO, where quantitative observations of utility are available at each step. Before discussing the case with preference feedback, we briefly review kernel ridge regression in the standard BO setting.

Hypothetically, assume a dataset $\{(x_i, o_i)\}_{i=1}^{t}$ of observations of $f$ is available, where $o_i = f(x_i) + \varepsilon_i$, with observation noise $\varepsilon_i$. Kernel ridge regression would provide a powerful predictor and uncertainty estimate of $f$, as follows:

$$\hat{f}_t(x) = k_t^\top(x)(K_t + \lambda I)^{-1} \boldsymbol{o}_t$$
$$\hat{\sigma}_t^2(x) = k(x, x) - k_t^\top(x)(K_t + \lambda I)^{-1} k_t(x), \quad (5)$$

where $k_t(x) = [k(x, x_i)]_{i=1}^{t}$ represents the pairwise kernel values between the prediction point $x$ and the observation points, $K_t = [k(x_i, x_j)]_{i,j=1}^{t}$ is the kernel (or covariance) matrix, $\lambda > 0$ is a free parameter, and $\boldsymbol{o}_t = [o_i]_{i=1}^{t}$ is the vector of observation values. The prediction function $\hat{f}_t$ here is the solution to the following regularized least squares

error optimization (see, e.g., Schölkopf et al., 2001):

$$\hat{f}_t = \arg\min_{g \in \mathcal{H}_k} \sum_{i=1}^{t} (g(z_i) - o_i)^2 + \frac{\lambda}{2} \|g\|_{\mathcal{H}_k}^2, \qquad (6)$$

where $\lambda$ is the same parameter as in (5). Confidence intervals of the form $|f(z) - \hat{f}_t(z)| \leq \hat{\beta}(\delta)\hat{\sigma}_t(z)$, where $\hat{\beta}(\delta)$ is a confidence interval width multiplier for a $1 - \delta$ confidence level, have been shown in several works (Abbasi-Yadkori, 2013; Chowdhury & Gopalan, 2017; Vakili et al., 2021a; Whitehouse et al., 2024) under various assumptions, and serve as key building blocks in the analysis and algorithm design of standard BO.

In the absence of straightforward observations $o_t$ and with preference-based feedback, a closed-form prediction is no longer available. Intuitively, this case resembles a classification-like problem with binary outputs, where we can employ a logistic negative log-likelihood loss. Specifically, for a history of preference feedback $\mathbb{H}_t = (x_1, x_1', y_1), \ldots, (x_t, x_t', y_t)$ in the BOHF framework, we define the following loss:

$$\mathcal{L}_k(h, \mathbb{H}_t) = \sum_{i=1}^{t} -y_i \log \mu(h(x_i, x_i'))$$
$$- (1 - y_i) \log(1 - \mu(h(x_i, x_i')) + \frac{\lambda}{2} \|h\|_{\mathcal{H}_k}^2$$

A prediction $h_t$ of the preference function $h$ (difference in the utilities) can be obtained as:

$$h_t = \arg\min_{h \in \mathcal{H}_k} \mathcal{L}_k(h, \mathbb{H}_t), \qquad (7)$$

which represents the minimizer of the regularized negative log-likelihood loss.

To solve this minimization problem, we apply the Representer Theorem, similar to Pásztor et al. (2024), which provides a parametric representation of $h_t$:

$$h_t(\cdot) = \sum_{i=1}^{t} \theta_i \mathbb{k} \left(\cdot, (x_i, x_i')\right), \qquad (8)$$

in terms of $\boldsymbol{\theta}_t = [\theta_1, \theta_2, \cdots, \theta_t]^\top \in \mathbb{R}^t$. With a slight abuse of notation, replacing $h$ with $\boldsymbol{\theta}$ in $\mathcal{L}_k$, the regularized negative log-likelihood loss can then be rewritten in terms of the parameter vector $\boldsymbol{\theta}$ as follows:

$$\mathcal{L}_k(\boldsymbol{\theta}, \mathbb{H}_t) = \sum_{i=1}^{t} -y_i \log \mu(\boldsymbol{\theta}^\top \mathbb{k}_t(x_i, x_i'))$$
$$- (1 - y_i) \log(1 - \mu(\boldsymbol{\theta}^\top \mathbb{k}_t(x_i, x_i')) + \frac{\lambda}{2} \|\boldsymbol{\theta}\|_2^2, \qquad (9)$$

where $\mathbb{k}_t(z) = [\mathbb{k}(z, (x_j, x_j'))]_{j=1}^{t}$ is the kernel values between the pair $z$ and observation pairs.

Similar to (5), we have an uncertainty estimation for each $z \in \mathcal{X} \times \mathcal{X}$ as follows

$$\sigma_t^2(z) = \mathbb{k}(z, z) - \mathbb{k}_t^\top(z)(\mathbb{K}_t + \lambda\kappa I)^{-1}\mathbb{k}_t(z), \qquad (10)$$

where the notation $\mathbb{K}_t = [\mathbb{k}\left((x_i, x_i'), (x_j, x_j')\right)]_{i,j=1}^{t}$ represents the (dueling) kernel matrix on the space of pair observations $\mathcal{X} \times \mathcal{X}$. Note the subtle difference in the definition of $\sigma_t^2$ above for the preference-based feedback case compared to the conventional kernel-based regression case, where the free parameter $\lambda$ is multiplied by $\kappa$, reflecting the effect of the sigmoid nonlinearity on the quality of prediction.

Centered around the prediction $\mu(h_t(\cdot))$ and incorporating the uncertainty estimate from kernel ridge regression, as defined in Equation (10), we can construct $1 - \delta$ confidence intervals of the form:

$$|\mu(h_t(z)) - \mu(h(z))| \leq \beta_t(\delta)\sigma_t(z),$$

for a pair of interest $z = (x, x')$. In Theorem 4.7, we prove a novel confidence interval of this form applicable to the analysis of our algorithm.

## 3. Algorithm Description

In this section, we present the Multi-Round Learning from Preference-based Feedback (MR-LPF) algorithm, inspired by Li & Scarlett (2022), designed to achieve low regret within the BOHF framework described in Section 2.1.

The algorithm partitions the time horizon $T$ into $R$ rounds, indexed by $r = 1, 2, \ldots, R$. During each round $r$, a total of $N_r$ samples are collected, ensuring that the cumulative number of samples across all rounds equals $T$, i.e., $\sum_{r=1}^{R} N_r = T$. We define $t_r = \sum_{j=1}^{r} N_j$ as the time step at the end of round $r$. The size of each round is determined as follows: $N_1 = \lceil\sqrt{T}\rceil$, $N_r = \lceil\sqrt{N_{r-1}T}\rceil$ for $1 < r < R$, and $N_R = \min\{\lceil\sqrt{N_{R-1}T}\rceil, T - t_{R-1}\}$.

We introduce the notations $\sigma_{(n,r)}(x, x')$ and $h_{(n,r)}(x, x')$ to represent the kernel-based uncertainty estimate and prediction, respectively, from the first $n$ samples in round $r$ according to Section 2.3.

MR-LPF maintains a set $\mathcal{M}_r$ of actions in each round that are likely to be the most preferable. Initially, $\mathcal{M}_1$ is set to $\mathcal{X}$ and is updated at the end of each round while satisfying a nested structure, $\mathcal{M}_r \subseteq \mathcal{M}_{r-1}$, as subsequently described.

Within each round $r$, the $n$-th sample is chosen as the pair of actions within $\mathcal{M}_r$ that maximizes uncertainty :

$$(x_{(n,r)}, x_{(n,r)}') = \arg\max_{x, x' \in \mathcal{M}_r} \sigma_{(n-1,r)}(x, x'). \qquad (11)$$

The preference feedback for this pair $y_{(n,r)} = \mathbb{1}\{x_{(n,r)} \succ x_{(n,r)}'\}$ is then revealed to the algorithm. The tuple $(x_{(n,r)}, x_{(n,r)}', y_{(n,r)})$ is added to the observations specific

to round $r$: $\mathbb{H}_{n,r} = \mathbb{H}_{n-1,r} \cup \{(x_{(n,r)}, x'_{(n,r)}, y_{(n,r)})\}$, which is initialized as an empty set at the beginning of the round: $\mathbb{H}_{0,r} = \emptyset$.

At the end of round $r$, we compute the prediction function $h_{(N_r,r)}$ based on observations $\mathbb{H}_{N_r,r}$, following the method of minimizing the regularized negative log-likelihood loss described in Section 2.3. Subsequently, we update $\mathcal{M}_r$ according to the following rule:

$$\mathcal{M}_{r+1} = \{x \in \mathcal{M}_r | \forall x' \in \mathcal{M}_r :$$
$$\mu(h_{(N_r,r)}(x, x')) + \beta_{(r)}\sigma_{(N_r,r)}(x, x') \geq 0.5\}. \tag{12}$$

The round specific parameters $\beta_{(r)}$ are designed in a way that the left hand side of the inequality is an upper confidence bound on the probability of favoring $x$ over $x'$ (the values are given in Theorem 4.1). The rationale here is that when an upper confidence bound on the probability of preferring $x$ to any $x'$ is greater than $0.5$, $x$ is plausible to be the most preferred action. Therefore, we keep it in the update of $\mathcal{M}_{r+1}$. All other actions are removed as they are unlikely to be the most preferred. More precisely, as we will show in the analysis, with high probability, the removed actions are not the most preferred, while the most preferred actions remain within the sets $\mathcal{M}_r$ and are not removed. A pseudocode is provided in Algorithm 1.

When the confidence intervals shrink at a sufficiently fast rate, only near-optimal actions remain in $\mathcal{M}_r$ as the rounds progress. This is a key aspect of our algorithm's design, which eliminates the dependency of regret scaling on $\kappa$ by ensuring that the algorithm gradually queries only closely preferred actions. Recall the discussion following Equation (2). In the next section, we provide an analysis of the performance guarantees of the algorithm.

## 4. Analysis of MR-LPF

In this section, we present our main results on the performance of MR-LPF (Algorithm 1). The performance is given in terms of the maximum information gain defined as

$$\Gamma_\lambda(T) = \max_{(x_1,x_1'),\dots(x_T,x_T')} \frac{1}{2} \log\det\left(I + \lambda^{-1}\mathbb{K}_T\right), \tag{13}$$

where $\mathbb{K}_T$ is the kernel matrix of $T$ observations.[2]

**Theorem 4.1** (Regret bound for MR-LPF)**.** *Consider the BOHF framework described in Section 2.1 and the MR-LPF algorithm presented in Algorithm 1. For $\delta \in (0,1)$, in MR-LPF, let*

$$\beta_{(r)}(\delta) = L\left(B + \sqrt{\frac{\kappa_r}{\lambda}\log(\frac{2R|\mathcal{X}|}{\delta})}\right), \tag{14}$$

---

[2]Unlike in Section 1, where $\lambda$ was omitted from the expression of $\Gamma$, we include it here for clarity.

---

**Algorithm 1** MR-LPF

---
**Require:** $\forall r, \beta_{(r)}$; time horizon $T$
  $\mathcal{M}_1 \leftarrow \mathcal{X}, t \leftarrow 1$
  **for** $r = 1, 2, \cdots, R$ **do**
    Initialize $\mathbb{H}_{0,r} = \{\}$
    **for** $n = 1, 2, \cdots, N_r$ **do**
      Select the pair of actions $(x_{(n,r)}, x'_{(n,r)})$ that maximizes the variance, with ties broken arbitrarily:
      $(x_{(n,r)}, x'_{(n,r)}) = \arg\max_{x,x'\in\mathcal{M}_r} \sigma_{(n-1,r)}(x, x')$
      $t \leftarrow t + 1$
      **if** $t \geq T$ **then**
        Terminate
      **end if**
      Observe $y_{(n,r)} = \mathbb{1}\{x_{(n,r)} \succ x'_{(n,r)}\}$
      $\mathbb{H}_{n,r} = \mathbb{H}_{n-1,r} \cup \{(x_{(n,r)}, x'_{(n,r)}, y_{(n,r)})\}$
    **end for**
    Update $h_{(N_r,r)}$ based on observations in $\mathbb{H}_{N_r,r}$
    Update the set of maximizers $\mathcal{M}_{r+1}$ by removing actions unlikely to be optimal:
    $\mathcal{M}_{r+1} = \{x \in \mathcal{M}_r | \forall x' \in \mathcal{M}_r : \mu(h_{(N_r,r)}(x, x')) + \beta_{(r)}\sigma_{(N_r,r)}(x, x') \geq 0.5\}$
  **end for**

---

*where, $B$ is the upper bound on the RKHS norm of $f$ given in Assumption 2.1, $L = \max_{x,x'\in\mathcal{X}} \dot\mu(h(x, x'))$, $\kappa_1 = \kappa$ defined in Equation (2), $\forall r > 1, \kappa_r = 6$, $\lambda$ is the regularization parameter of the kernel-based regression. Then, for some constant $T_0 > 0$, independent of $T$ (specified in Appendix B), and for all $T \geq T_0$, with probability at least $1 - \delta$:*

$$R(T) \leq 2CR\beta_{(R)}(\delta)\sqrt{\Gamma_{(4\lambda)}(T)}\left(T^{1/2} + 1\right),$$

*where $R \leq \lceil \log_2 \log_2(T)\rceil + 1$ is the maximum number of rounds and $C = 2\sqrt{\frac{2}{\log(1+4(6\lambda)^{-1})}}$ is a constant.*

*Remark* 4.2. The value of $\Gamma_\lambda(T)$ is kernel-specific and algorithm-independent. This term is a common complexity measure that appears in the analysis of both BO and BOHF in the existing literature (e.g., see Srinivas et al., 2010; Pásztor et al., 2024; Xu et al., 2024). Bounds on $\Gamma_\lambda(T)$ have been established for various kernels. In particular, for linear kernels, $\Gamma_\lambda(T) = \mathcal{O}(d\log(T))$. For kernels with exponentially decaying Mercer eigenvalues, such as the Squared Exponential (SE) kernel, $\Gamma_\lambda(T) = \mathcal{O}(\text{poly}\log(T))$. For kernels with polynomially decaying eigenvalues, $\Gamma_\lambda(T)$ grows polynomially (though sublinearly) with $T$. For example, in the case of the Matérn family of kernels, $\Gamma_\lambda(T) = \tilde{\mathcal{O}}(T^{\frac{d}{2\nu+d}})$, where $d$ is the input dimension and $\nu > 0.5$ is the smoothness parameter (see, e.g., Vakili et al., 2021b). In Proposition 4 of Pásztor et al. (2024), it is shown that the eigenvalues of the dueling kernel $\mathbb{k}$ are exactly twice those of the original kernel $k$ (see their Appendix C.1). Since the

maximum information gain $\Gamma_\lambda(T)$ scales with the decay rate of the kernel eigenvalues (Vakili et al., 2021b), both kernels exhibit the same scaling of the information gain with $T$.

*Remark* 4.3. By substituting the value of $\beta_{(R)}(\delta)$, the expression of the regret bound can be simplified to

$$R(T) = \tilde{\mathcal{O}}\left(\sqrt{\Gamma_\lambda(T)T\log(\frac{|\mathcal{X}|}{\delta})}\right), \qquad (15)$$

as $T$ becomes large. This represents a sublinear regret growth rate for a broad class of commonly used kernels where $\Gamma_\lambda(T)$ grows sublinearly with $T$.

Our regret bounds eliminate the dependency on $\kappa$. MR-LPF gradually queries only closely preferred actions, reducing the effective impact of the curvature of the link function. Our regret bounds also show an $\mathcal{O}\left(\sqrt{\Gamma(T)}\right)$ improvement compared to Pásztor et al. (2024) and an $\mathcal{O}\left((\Gamma(T)T)^{1/4}\right)$ improvement over Xu et al. (2024). This becomes particularly crucial for kernels with polynomially decaying eigenvalues, where existing results may become vacuous, failing to guarantee sublinear regret in $T$.

### 4.1. Sample Complexity and Simple Regret

In certain applications, the learner may be primarily concerned with eventual performance, specifically the simple regret after $T$ observations. Accordingly, we can pose the dual question: *How many samples are required to achieve $\epsilon$ simple regret?* This aspect of our algorithm's performance is formalized in the following corollary.

**Corollary 4.4.** *Under the setting of Theorem 4.1, assume $T = t_R$, the time step at the end of round $R$. For any action $\hat{x}_T \in \mathcal{M}_{R+1}$, we have, with probability at least $1 - \delta$,*

$$\mathbb{P}(x^\star \succ \hat{x}_T) - \frac{1}{2} \leq 2\beta_{(R)}(\delta)C\sqrt{\frac{R\Gamma_{(4\lambda)}(T)}{T}}. \qquad (16)$$

The proof is given in Appendix B, that follows from Theorem 4.1.

**Corollary 4.5.** *As a consequence of Corollary 4.4, assume we run MR-LPF for $T = t_R$ rounds and select $\hat{x}_T \in \mathcal{M}_{R+1}$ arbitrarily. In the case of a linear kernel with some $T = \tilde{\mathcal{O}}\left(\frac{d\log(\frac{1}{\delta})}{\epsilon^2}\right)$, an SE kernel with some $T = \tilde{\mathcal{O}}\left(\frac{\log(\frac{1}{\delta})}{\epsilon^2}\right)$, and a Matérn kernel with some $T = \tilde{\mathcal{O}}\left(\frac{\log(\frac{1}{\delta})}{\epsilon^{2+\frac{d}{\nu}}}\right)$, with probability at least $1 - \delta$, at most $\epsilon$ error is guaranteed: $P(x^\star \succ \hat{x}_T) - \frac{1}{2} \leq \epsilon$.*

*Remark* 4.6. Our sample complexities match the $\Omega\left(\frac{1}{\epsilon^{2+\frac{d}{\nu}}}\right)$ lower bounds for conventional BO with Matérn kernels, as established in Scarlett et al. (2017) (up to logarithmic terms).

These bounds apply to scalar-valued feedback, which is richer than the binary preference feedback used in BOHF.

For technical details, consider a standard BO setting with scalar observations $o_i = f(x_i) + \varepsilon_i$, where $\varepsilon_i$ are i.i.d., zero-mean noise terms (following the notation in Section 2.3). Suppose that at each step $t$, instead of observing $o_t = f(x_t) + \varepsilon_t$ and $o'_t = f(x'_t) + \varepsilon'_t$, we receive binary preference feedback $y_t = \mathbb{1}\{o_t > o'_t\}$. Under the BTL model, this corresponds to the case where the noise difference $\varepsilon'_t - \varepsilon_t$ follows a logistic distribution, which can arise if the individual noise terms $\varepsilon_t$ are Gumbel-distributed. Thus, the lower bound on sample complexity in the BOHF setting should be at least half of that of conventional BO under Gumbel noise for achieving at most $\epsilon$ loss in the value of the target function.

Since the lower bound construction in Scarlett et al. (2017) assumes Gaussian noise, a formal comparison is not strictly valid (as the BTL model corresponds to Gumbel noise). We therefore present this connection as an informal justification of the tightness of our bounds, rather than a formal optimality proof.

### 4.2. Confidence Intervals and Proofs

An important building block in analyzing the performance of MR-LPF is the confidence intervals applied to the samples collected in each round. We now present a formal statement of this result.

**Theorem 4.7** (Confidence Bounds). *Consider the kernel-based prediction $h_t$ and uncertainty estimate $\sigma_t$ for a dataset $\mathbb{H}_t$ and a known kernel $\Bbbk$, as given in Equations (7) and (10) satisfying Assumption 2.1. Assume the observation points $\{(x_i, x'_i)\}_{i=1}^t$ are independent of the observation values $\{y_i\}_{i=1}^t$. For a fixed $(x, x') \in \mathcal{X} \times \mathcal{X}$ and for any $\delta \in (0, 1)$, we have, with probability at least $1 - \delta$,*

$$|\mu(h_t(x, x')) - \mu(h(x, x'))| \leq \beta(\delta)\sigma_t(x, x'), \qquad (17)$$

*where $\beta(\delta) = L\left(B + \frac{1}{2}\sqrt{\frac{2\kappa}{\lambda}\log(2/\delta)}\right)$, $L = \sup_{x,x' \in \mathcal{X}} \dot{\mu}(h(x, x'))$ as defined in Theorem 4.1, $B$ is the RKHS norm bound specified in Assumption 2.1, $\lambda$ is the parameter in kernel-based regression, and $\kappa$ is defined in Equation (2).*

A key distinction in our results is that our confidence interval is tighter than the one presented in Pásztor et al. (2024) by a factor of $\mathcal{O}(\sqrt{\Gamma(T)})$. This improvement comes from the multi-round structure and action selection rule within each round of the algorithm, which ensures that the observation points used for confidence intervals at the end of rounds are independent of the observation values within that round. This removes certain intricate dependencies in deriving the confidence interval. Recall that the observation points in

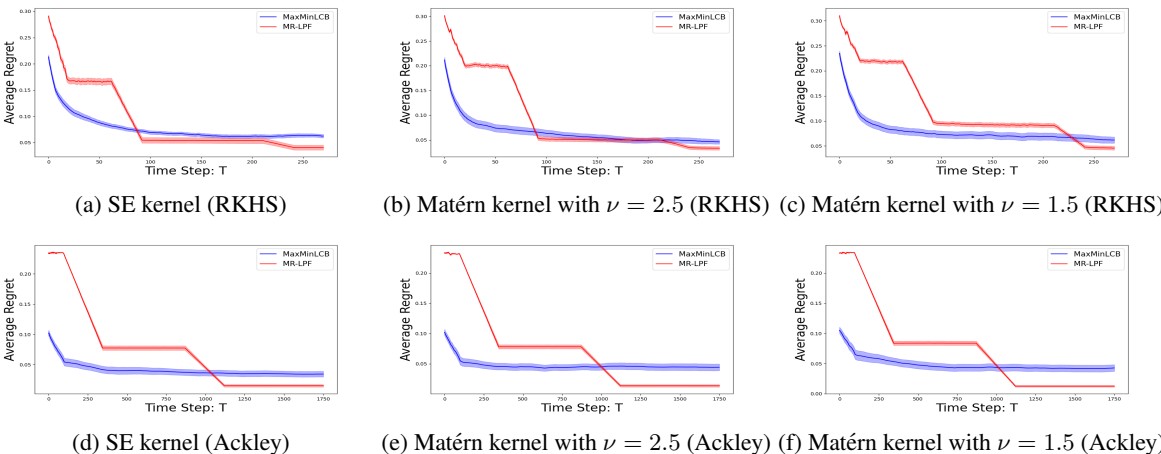

(a) SE kernel (RKHS)       (b) Matérn kernel with $\nu = 2.5$ (RKHS)   (c) Matérn kernel with $\nu = 1.5$ (RKHS)

(d) SE kernel (Ackley)      (e) Matérn kernel with $\nu = 2.5$ (Ackley)   (f) Matérn kernel with $\nu = 1.5$ (Ackley)

*Figure 1.* Average Regret against $T$ with RKHS test functions (top row) and Ackley test function (bottom row). The shaded area represents the standard error.

each round, $(x_{(n,r)}, x'_{(n,r)})$, are collected solely based on the variance, which is independent of the observation values by definition. In contrast, both the MaxMinLCB algorithm in Pásztor et al. (2024) and the POP-BO algorithm in Xu et al. (2024) select observation point at step $t$ based on statistics that depend on $\{y_i\}_{i=1}^{t-1}$. We emphasize that our algorithm is by no means a pure exploration algorithm; it effectively balances exploration and exploitation by learning and updating $\mathcal{M}_r$ at the end of each round.

Given the confidence intervals in Theorem 4.7, the update rule of $\mathcal{M}_r$ in MR-LPF ensures that the best action is not eliminated (Lemma B.2). Additionally, we can use the confidence intervals to bound the regret for each action in $\mathcal{M}_r$, based on the maximum variance in predictions from previous rounds. By summing up the regret over all rounds, we achieve the overall regret bound, with details provided in Appendix B. For proof of Theorem 4.7, see Appendix C.

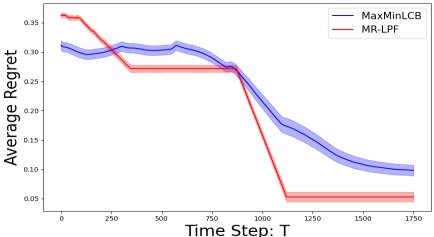

*Figure 2.* Average regret against $T$ for the experiment with Yelp Open Dataset. The shaded area represents the standard error.

## 5. Experiments

We run numerical experiments to evaluate the performance of MR-LPF and compare it to MaxMinLCB (Pásztor et al.,

2024, Algorithm 1) on various test functions, including both synthetic and real-world cases. Our implementation is publicly available.[3]

We first select the test function $f$ as an arbitrary function in the RKHS of a known kernel. To do this, we choose 10 points in the $[0, 1]$ interval and assign them random values. We then fit a standard kernel ridge regression to these samples using a kernel $k$ and use its mean as $f$. The kernel $k$ is set to the SE kernel and Matérn kernels with smoothness parameters $\nu = 2.5$ and $\nu = 1.5$. This is a common approach to constructing functions in an RKHS (see, e.g., Chowdhury & Gopalan, 2017). We also test the algorithms on the Ackley function, similar to Pásztor et al. (2024). The Ackley function has a diverse optimization landscape, featuring multiple local minima, flat plateaus, and valleys, making it a popular choice in non-convex optimization literature (Jamil & Yang, 2013).

To showcase the utility of our approach in real-world applications, we experimented using the Yelp Open Dataset[4] of restaurant reviews. This serves as a proof of concept, demonstrating both the potential integration of BOHF with LLM-generated vector embeddings and the scalability of the method to higher-dimensional domains. The objective is to learn user preferences from comparative feedback and recommend restaurants tailored to each user's choices. After data filtering and pre-processing, the dataset consists of 275 restaurants, 20 users, and 2563 reviews. Each restaurant is represented by a 32-dimensional vector embedding of its text-based reviews, generated using OpenAI's text-

---

[3]https://github.com/ayakayal/BOHF_code_submission

[4]Yelp Open Dataset

embedding-3-large model[5]. Users rate restaurants on a scale from 1 to 5. We adopt the experimental setup and Yelp data preprocessing from Pásztor et al. (2024) to ensure a fair evaluation. While we implemented our own version instead of using their code[6] directly, we acknowledge their contribution in establishing this benchmark, which inspired our experiment. We frame this problem within the BOHF framework, where the action set $\mathcal{X}$ consists of 275 restaurants, each represented as a 32-dimensional vector, and the utility values $f$ correspond to user ratings. SE kernel is used for these experiments. For details on the experimental setup, see Appendix D.

We plot the average regret at each time step, averaged over 60 independent runs. Figure 1 shows the results on the RKHS and Ackley test functions, while Figure 2 presents the results on the Yelp Open Dataset. MR-LPF consistently achieves lower regret than MaxMinLCB across all test functions. The initial regret of MR-LPF reflects highly exploratory behavior during the early rounds. At the end of each round $r$, suboptimal actions are removed from $\mathcal{M}_r$, leading to the sharp drops that eventually result in near-optimal actions in later rounds. Relatively constant behavior within rounds represents exploration, while sharp drops indicate exploitation.

## 6. Conclusion

We proposed MR-LPF for the BOHF problem and proved regret bounds and sample complexities, significantly improving upon existing work. We established that the number of preferential feedback samples required to identify near-optimal actions is of the same order as the number of scalar-valued feedback samples. Numerical experiments on both synthetic and real-world examples support our analytical results.

## Acknowledgments

We thank the reviewers and the Area Chair of ICML for their valuable feedback. We are especially grateful for the Area Chair's thorough and insightful comments, which clearly reflect a significant investment of time and have substantially improved the final version of this work.

## Impact Statement

This work presents analytical research aimed at advancing the field of Machine Learning. While the work has potential societal implications, none are considered immediate or require specific emphasis at this time.

---

[5] OpenAI Vector Embeddings
[6] https://github.com/lasgroup/MaxMinLCB.

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

## A. RKHS and Mercer Theorem

Mercer's theorem (Mercer, 1909) provides a way to represent a kernel using an infinite-dimensional feature map (see, e.g., Christmann & Steinwart, 2008, Theorem 4.49). Let $\mathcal{Z}$ be a compact metric space, and let $\nu$ be a finite Borel measure on $\mathcal{Z}$ (in this context, we consider the Lebesgue measure in a Euclidean space). Denote by $L^2_\nu(\mathcal{Z})$ the set of square-integrable functions on $\mathcal{Z}$ with respect to $\nu$. Additionally, we say that a kernel is square-integrable if

$$\int_{\mathcal{Z}} \int_{\mathcal{Z}} k^2(z, z') \, d\nu(z) d\nu(z') < \infty.$$

**Theorem A.1.** *(Mercer Theorem) Let $\mathcal{Z}$ be a compact metric space and $\nu$ be a finite Borel measure on $\mathcal{Z}$. Let $k$ be a continuous and square-integrable kernel, inducing an integral operator $T_k : L^2_\nu(\mathcal{Z}) \to L^2_\nu(\mathcal{Z})$ defined by*

$$(T_k f)(\cdot) = \int_{\mathcal{Z}} k(\cdot, z') f(z') \, d\nu(z'),$$

*where $f \in L^2_\nu(\mathcal{Z})$. Then, there exists a sequence of eigenvalue-eigenfeature pairs $\{(\gamma_m, \varphi_m)\}_{m=1}^\infty$ such that $\gamma_m > 0$, and $T_k \varphi_m = \gamma_m \varphi_m$, for $m \geq 1$. Moreover, the kernel function can be represented as*

$$k(z, z') = \sum_{m=1}^\infty \gamma_m \varphi_m(z) \varphi_m(z'),$$

*where the convergence of the series holds uniformly on $\mathcal{Z} \times \mathcal{Z}$.*

According to the Mercer representation theorem (e.g., see, Christmann & Steinwart, 2008, Theorem 4.51), the RKHS induced by $k$ can consequently be represented in terms of $\{(\gamma_m, \varphi_m)\}_{m=1}^\infty$.

**Theorem A.2.** *(Mercer Representation Theorem) Let $\{(\gamma_m, \varphi_m)\}_{i=1}^\infty$ be the Mercer eigenvalue-eigenfeature pairs. Then, the RKHS of $k$ is given by*

$$\mathcal{H}_k = \left\{ f(\cdot) = \sum_{m=1}^\infty w_m \gamma_m^{\frac{1}{2}} \varphi_m(\cdot) : w_m \in \mathbb{R}, \|f\|_{\mathcal{H}_k}^2 := \sum_{m=1}^\infty w_m^2 < \infty \right\}.$$

Mercer representation theorem indicates that the scaled eigenfeatures $\{\sqrt{\gamma_m} \varphi_m\}_{m=1}^\infty$ form an orthonormal basis for $\mathcal{H}_k$.

**Definition A.3.** A kernel $k$ is said to have a polynomial (exponential) eigendecay if $\gamma_m = \mathcal{O}(m^{-p})$ $(\gamma_m = \mathcal{O}(c^m))$, for some $p > 1$ $(c < 1)$, where $\gamma_m$ are the Mercer eigenvalues in decreasing order.

**Specific kernel functions:**

1. **Linear kernel:** $k(x, x') = x^T x'$

2. **Squared Exponential (SE) kernel:** $k(x, x') = \sigma^2 \exp\left(-\frac{|x-x'|^2}{2l^2}\right)$ where $\sigma^2$ is a scalar and $l > 0$ is referred to as the length-scale of the kernel.

3. **Matérn kernel:** $k(x, x') = \frac{2^{1-\nu}}{\Gamma(\nu)} \left(\sqrt{2\nu} \frac{|x-x'|}{l}\right)^\nu K_\nu\left(\sqrt{2\nu} \frac{|x-x'|}{l}\right)$ where $\nu > 0.5$ is the smoothness parameter of the kernel, $l$ is referred to as the length-scale, $K_\nu$ is the modified Bessel function, and $\Gamma$ is the Gamma function.

   For the Matérn kernel, the eigenvalues decay polynomially with $p = 1 + \frac{2\nu}{d}$ where $d$ is the input dimension.

## B. Proof of The Regret Bound and Sample Complexities

In this section, we provide a detailed proof of Theorem 4.1 on the regret bound of MR-LPF and following corollaries.

### B.1. Proof of Theorem 4.1

To prove this theorem, we bound the regret for each round and then sum these bounds over all the rounds.

**Regret in the first round:** The first round consists of $N_1 = \lceil \sqrt{T} \rceil$ samples. We note that for all $t$,

$$\frac{\mathbb{P}(x^\star \succ x_t) + \mathbb{P}(x^\star \succ x_t') - 1}{2} \leq \frac{1}{2}. \tag{18}$$

Consequently, the regret incurred in the first round in bounded by $\frac{1}{2}\lceil \sqrt{T} \rceil$.

For the second round onwards ($r \geq 2$), we introduce some notation and preliminaries that will assist in bounding the regret.

**High probability events:** Let us define the event $\mathcal{E}_r$ as the event that all the confidence intervals used in the round $r$ of the MR-LPF algorithm hold true. Specifically,

$$\mathcal{E}_r = \left\{ \forall x, x' \in \mathcal{M}_r : \left| \mu(h_{(N_r,r)}(x, x')) - \mu(h(x, x')) \right| \leq \beta_{(r)}(\delta)\sigma_{(N_r,r)}(x, x') \right\} \tag{19}$$

Recall that $\beta_{(r)}(\delta) = L\left( B + \sqrt{\frac{\kappa_r}{\lambda} \log(\frac{2R|\mathcal{X}|}{\delta})} \right)$. We also define $\mathcal{E} = \bigcup_{r=1}^R \mathcal{E}_r$.

**Sum of the posterior variances for a sequence of observations:** We apply the following bound on the sum of posterior variances in each round (see, e.g., Pásztor et al., 2024, Lemma 14).

$$\sum_{n=1}^{N_r} \sigma_{(n-1,r)}^2(x_{(n,r)}, x_{(n,r)}') \leq \frac{8}{\log(1 + 4(\lambda\kappa_r)^{-1})} \Gamma_{(\lambda\kappa_r)}(N_r). \tag{20}$$

By the selection rule of $(x_{(n,r)}, x_{(n,r)}')$ in MR-LPF as the points with the highest variance, we have that $\forall x, x' \in \mathcal{M}_r$, and $\forall n \leq N_r, \sigma_{(N_r,r)}(x, x') \leq \sigma_{(n-1,r)}(x_{(n,r)}, x_{(n,r)}')$. Combining this with Equation (20), we conclude that $\forall x, x' \in \mathcal{M}_r$,

$$\sigma_{(N_r,r)}(x, x') \leq \sqrt{\frac{8}{\log(1 + 4(\lambda\kappa_r)^{-1})}} \sqrt{\frac{\Gamma_{(\lambda\kappa_r)}(N_r)}{N_r}}. \tag{21}$$

**The value of $\kappa_r, r \geq 2$:** Recall the update rule for $\mathcal{M}_r$ in MR-LPF:

$$\mathcal{M}_{r+1} = \left\{ x \in \mathcal{M}_r | \forall x' \in \mathcal{M}_r : \mu(h_{(N_r,r)}(x, x')) + \beta_{(r)}\sigma_{(N_r,r)}(x, x') \geq \frac{1}{2} \right\} \tag{22}$$

Assuming $\mathcal{E}_1$, for all $x, x' \in \mathcal{M}_2$, we have

$$\mu(h(x, x')) + 2\beta_{(1)}\sigma_{(N_1,1)}(x, x') \geq \mu(h_{(N_1,1)}(x, x')) + \beta_{(1)}\sigma_{(N_1,1)}(x, x')$$
$$\geq \frac{1}{2}, \tag{23}$$

where the first inequality holds under $\mathcal{E}_1$ and the second inequality is a consequence of the update rule. Similarly, we have

$$\mu(h(x', x)) + 2\beta_{(1)}\sigma_{(N_1,1)}(x', x) \geq \frac{1}{2}. \tag{24}$$

We note that $\forall x, x' \in \mathcal{X}, \mu(h(x', x)) = 1 - \mu(h(x, x'))$. Thus, Equation (24) implies that

$$\mu(h(x, x')) \leq \frac{1}{2} + 2\beta_{(1)}\sigma_{(N_1,1)}(x', x). \tag{25}$$

Combining with (23), we obtain that

$$-2\beta_{(1)}\sigma_{(N_1,1)}(x, x') \leq \mu(h(x, x')) - \frac{1}{2} \leq 2\beta_{(1)}\sigma_{(N_1,1)}(x', x). \tag{26}$$

We previously established a bound on the standard deviation at the end of rounds in (21). Applying this to the first round, with length $N_1 = \lceil \sqrt{T} \rceil$, we can bound $\mu(h(x, x'))$ for all $x, x' \in \mathcal{M}_2$ within the interval $[\frac{1}{4}, \frac{3}{4}]$ by ensuring $2\beta_{(1)}\sigma_{(N_1,1)}(x', x) \leq \frac{1}{4}$. Specifically, let $T_0$ be the smallest integer satisfying

$$2\beta_{(1)}(\delta)\sqrt{\frac{8}{\log(1+4(\lambda\kappa)^{-1})}}\sqrt{\frac{\Gamma_{(\lambda\kappa)}(\lceil\sqrt{T_0}\rceil)}{\lceil\sqrt{T_0}\rceil}} \leq \frac{1}{4}. \tag{27}$$

Then, for any $T \geq T_0$, for all $x, x' \in \mathcal{M}_2$, we have $\mu(h(x, x')) \in [\frac{1}{4}, \frac{3}{4}]$. Recall that the derivative of the sigmoid function is given by $\dot{\mu}(x) = \mu(\cdot)(1 - \mu(\cdot))$. Consequently, the inverse of the derivative of the sigmoid applied to $h$, for the values of $x, x' \in \mathcal{M}_2$, is bounded as follows. For all $x, x' \in \mathcal{M}_2$,

$$\frac{1}{\mu(h(x, x'))(1 - \mu(h(x, x')))} \leq \frac{16}{3} < 6. \tag{28}$$

Thus, we can use $\kappa_r = 6$ for all $r \geq 2$, maintaining the validity of the confidence intervals.

**Lemma B.1.** *For $T \geq T_0$ specified in Equation (27), we have $\mathbb{P}(\mathcal{E}) \leq 1 - \delta$.*

The proof follows from Theorem 4.7, a union bound over all action pairs and rounds, and the bound on $\kappa_r$ derived above. We condition the remainder of the proof on the event $T \geq T_0$ and $\mathcal{E}$.

**The best action $x^\star$ will not be removed.** Assuming $\mathcal{E}$, the best action will not be removed from the sets $\mathcal{M}_r$ by the MR-LPF algorithm in any round. We formalize this observation in the following lemma.

**Lemma B.2.** *Under event $\mathcal{E}$, $x^\star \in \mathcal{M}_R$.*

The proof follows from the observation that $\mu(h(x^\star, x)) \geq \frac{1}{2}$ for all $x$. Combining with the confidence intervals in $\mathcal{E}$, $\forall r$, $\forall x \in \mathcal{M}_r$, $\mu(h_{(N_r,r)}(x^\star, x)) + \beta_{(r)}(\delta)\sigma_{(N_r,r)}(x^\star, x) \geq \frac{1}{2}$. Consequently, the best action $x^\star$ will not be removed.

We are now ready to bound the regret in rounds $r \geq 2$.

**Regret bound in each round $r \geq 2$:** For each $x \in \mathcal{M}_r$, we use the update rule of $\mathcal{M}_r$ in MR-LPF to bound the regret with respect to the optimal action. Recall that in Lemma B.2, we showed that the optimal action remains in $\mathcal{M}_r$ for all $r$. We have

$$\mu(h(x, x^\star)) + 2\beta_{(r-1)}(\delta)\sigma_{(N_{r-1},r-1)}(x, x^\star) \geq \mu(h_{(N_{r-1},r-1)}(x, x^\star)) + \beta_{(r-1)}(\delta)\sigma_{(N_{r-1},r-1)}(x, x^\star)$$
$$\geq \frac{1}{2}, \tag{29}$$

where the first inequality holds under $\mathcal{E}$, and the second inequality follows from the update rule of $\mathcal{M}_r$. Then, we have

$$\mu(h(x^\star, x)) = 1 - \mu(h(x, x^\star))$$
$$\leq \frac{1}{2} + 2\beta_{(r-1)}(\delta)\sigma_{(N_{r-1},r-1)}(x, x^\star), \tag{30}$$

The equality follows from $\mu(-\cdot) = 1 - \mu(\cdot)$, and the inequality follows from (29).

We thus have for all $x \in \mathcal{M}_r$,

$$\mu(h(x^\star, x)) - \frac{1}{2} \leq 2\beta_{(r-1)}(\delta)\sigma_{(N_{r-1},r-1)}(x, x^\star)$$
$$\leq 2\beta_{(r-1)}(\delta)C\sqrt{\frac{\Gamma_{(\lambda\kappa_{r-1})}(N_{r-1})}{N_{r-1}}}, \tag{31}$$

where the second inequality is proven in (21), and we use $C = \sqrt{\frac{8}{\log(1+4(6\lambda)^{-1})}}$ to simplify the notation. This bound holds for all points in round $r$. Therefore, to obtain the regret in round $r$, it is sufficient to multiply this bound by $N_r$. This results in the following bound on the regret in round $r$:

$$\text{Regret in Round } r \leq 2\beta_{(r-1)}(\delta)CN_r\sqrt{\frac{\Gamma_{(\lambda\kappa_{r-1})}(N_{r-1})}{N_{r-1}}}$$

$$\leq 2\beta_{(r-1)}(\delta)C\left(\sqrt{T\Gamma_{(4\lambda)}(T)} + \frac{1}{\sqrt{N_{r-1}}}\sqrt{\Gamma_{(4\lambda)}(T)}\right)$$

$$\leq 2\beta_{(r-1)}(\delta)C\left(\sqrt{T\Gamma_{(4\lambda)}(T)} + T^{-1/4}\sqrt{\Gamma_{(4\lambda)}(T)}\right), \tag{32}$$

where the second inequality is obtained by substituting $N_r = \lceil\sqrt{N_{r-1}T}\rceil$ and using $\lceil\cdot\rceil \leq \cdot + 1$. We also use that $\Gamma_{(\lambda\kappa_{r-1})}(.) \leq \Gamma_{(4\lambda)}(.)$ since $\kappa_{r-1} \geq 4$. The third inequality follows from $N_r \geq \sqrt{T}$ for all $r \geq 1$.

**Total regret:** The number of rounds $R$ is at most $\lceil\log\log_2(T)\rceil + 1$ (Li & Scarlett, 2022, Proposition 1). Using the bound on regret in each round, we can bound the total regret of MR-LPF algorithm as follows

$$R(T) \leq 2CR\beta_{(R)}(\delta)\sqrt{T\Gamma_{(4\lambda)}(T)} + 2CR\beta_{(R)}(\delta)T^{-1/4}\sqrt{\Gamma_{(4\lambda)}(T)}. \tag{33}$$

This completes the proof of Theorem 4.1.

### B.2. Proof of Corollary 4.4

Since the size $N_r$ of rounds increase with $r$, we have $N_R \geq T/R$. In the proof of Theorem 4.1, in (31), we showed that, for all $x \in \mathcal{M}_r$

$$\mu(h(x^\star, x)) - \frac{1}{2} \leq 2\beta_{(r-1)}(\delta)C\sqrt{\frac{\Gamma_{(\lambda\kappa_{r-1})}(N_{r-1})}{N_{r-1}}}$$

Thus, for $x \in \mathcal{M}_{R+1}$, we have

$$\mu(h(x^\star, x)) - \frac{1}{2} \leq 2\beta_{(R)}(\delta)C\sqrt{\frac{\Gamma_{(4\lambda)}(N_R)}{N_R}}$$

$$\leq 2\beta_{(R)}(\delta)C\sqrt{\frac{R\Gamma_{(4\lambda)}(N_R)}{T}} \tag{34}$$

$$\leq 2\beta_{(R)}(\delta)C\sqrt{\frac{R\Gamma_{(4\lambda)}(T)}{T}}, \tag{35}$$

where, for the second inequality, we used $N_R \geq \frac{T}{R}$, and for the third inequality, we used $N_R \leq T$.

### B.3. Proof of Corollary 4.5

Following the bounds obtained in Corollary 4.4, we determine $T$ that ensures $\mu(h(\hat{x}_T, x)) - \frac{1}{2} \leq \epsilon$, after $T$ steps. For this, we need specification of $\Gamma_\lambda(T)$.

In the case of linear kernels, we have $\Gamma_\lambda(T) = \mathcal{O}(d\log(T))$. Consequently, a choice of $T = \tilde{\mathcal{O}}\left(\frac{d\log\left(\frac{1}{\delta}\right)}{\epsilon^2}\right)$ ensures $\mu(h(x^\star, x)) - \frac{1}{2} \leq \epsilon$.

In the case of SE kernel, we have $\Gamma_\lambda(T) = \mathcal{O}(\log^{d+1}(T))$. Consequently, a choice of $T = \tilde{\mathcal{O}}\left(\frac{\log\left(\frac{1}{\delta}\right)}{\epsilon^2}\right)$ ensures $\mu(h(x^\star, x)) - \frac{1}{2} \leq \epsilon$.

In the case of Matérn kernel, we have $\Gamma_\lambda(T) = \tilde{\mathcal{O}}(T^{\frac{d}{2\nu+d}})$. Consequently, a choice of $T = \tilde{\mathcal{O}}\left(\frac{\log\left(\frac{1}{\delta}\right)}{\epsilon^{2+\frac{d}{\nu}}}\right)$ ensures $\mu(h(x^\star, x)) - \frac{1}{2} \leq \epsilon$.

For the bound on $\Gamma_\lambda(T)$ see, e.g., Vakili et al. (2021b).

## C. Proof of Theorem 4.7

Recall the conventional kernel-based regression discussed in Section 2. Various confidence intervals of the form $|f(z) - \hat{f}_t(z)| \leq \hat{\beta}(\delta)\hat{\sigma}_t(z)$, where $\hat{f}_t(z)$ and $\hat{\sigma}_t(z)$ are the conventional prediction and standard deviation, and $\hat{\beta}(\delta)$ is a confidence interval width multiplier for a $1 - \delta$ confidence level, have been demonstrated in several works (Abbasi-Yadkori, 2013; Chowdhury & Gopalan, 2017; Vakili et al., 2021a; Whitehouse et al., 2024). As discussed in the preference-based case, the problem becomes more similar to a classification problem with binary feedback, and these confidence intervals are not directly applicable. Moreover, a closed-form solution for $h_t$ is not available, as it is only provided as the minimizer of the loss function given in Equation (7). Additionally, as discussed, this loss and its solution can be parameterized using the representer theorem.

$$\mathcal{L}_{\Bbbk}(\boldsymbol{\theta}, \mathbb{H}_t) = \sum_{i=1}^{t} -y_i \log \mu(\boldsymbol{\theta}^\top \Bbbk_t(x_i, x_i'))$$

$$- (1 - y_i) \log(1 - \mu(\boldsymbol{\theta}^\top \Bbbk_t(x_i, x_i')) + \frac{\lambda}{2}||\boldsymbol{\theta}||_2^2, \tag{36}$$

and

$$h_t(\cdot) = \sum_{i=1}^{t} \theta_i \Bbbk\left(\cdot, (x_i, x_i')\right). \tag{37}$$

For the remainder of the proof, and for simplicity of presentation, we use the notation $z = (x, x')$ and similarly $z_i = (x_i, x_i')$.

In both Xu et al. (2020b) and Pásztor et al. (2024), confidence intervals for $|h(z) - h_t(z)|$ are derived, with Pásztor et al. (2024) establishing tighter bounds. Their confidence intervals are based on the results of Faury et al. (2020) for logistic bandits and Whitehouse et al. (2024) for confidence intervals in kernel bandits. In comparison, our confidence intervals are tighter than those presented in Pásztor et al. (2024) by a factor of $\mathcal{O}(\sqrt{\Gamma_\lambda(T)})$. We achieve this improvement by assuming that the sequence of observation points $\{z_i\}_{i=1}^t$ is independent of the observation values $\{y_i\}_{i=1}^t$, inspired by Vakili et al. (2021a). This assumption is made possible in our work due to the design of the MR-LPF algorithm, where within each round, the observation points are selected based solely on kernel-based variance, which, by definition, does not depend on the observation values.

The main steps of the proof are similar to those in the proof of the confidence interval in Pásztor et al. (2024), and we will highlight the key differences in our proof. The key idea is that the derivative of the loss $\mathcal{L}_{\Bbbk}$, as given in Equation (36), is the null operator at the minimizer of the loss:

$$\nabla \mathcal{L}(\boldsymbol{\theta}_t, \mathbb{H}_t) = \sum_{i=1}^{t} -y_i \Bbbk(z_i, \cdot) + g_t(\boldsymbol{\theta}_t) = 0, \tag{38}$$

where $g_t(\boldsymbol{\theta}) : \mathcal{H}_{\Bbbk} \to \mathcal{H}_{\Bbbk}$ is a linear operator defined as

$$g_t(\boldsymbol{\theta}) = \sum_{i=1}^{t} \Bbbk(z_i, \cdot)\mu(\boldsymbol{\theta}^\top \Bbbk(z_i, \cdot)) + \lambda\boldsymbol{\theta}. \tag{39}$$

Recall that $\boldsymbol{\theta}_t$ is the minimizer of the loss in Equation (36). Consequently, we have $g_t(\boldsymbol{\theta}_t) = \sum_{i=1}^{t} y_i \Bbbk(z_i, \cdot)$.

Then, confidence intervals are proven for the gradient and extended to the preference function itself. We now introduce some auxiliary notation that will be helpful throughout the rest of the proof. Let $\boldsymbol{\Phi}_t = [\Bbbk(z_1, \cdot), \Bbbk(z_2, \cdot), \ldots, \Bbbk(z_t, \cdot)]^\top$, from which we define the kernel matrix $\mathbb{K}_t = \boldsymbol{\Phi}_t \boldsymbol{\Phi}_t^\top$ and the operator $S_t = \boldsymbol{\Phi}_t^\top \boldsymbol{\Phi}_t$. We also use $I_t$ to denote the $t$-dimensional identity matrix and $I_{\mathcal{H}}$ to denote the identity operator in the RKHS. Finally, we define $V_t = S_t + \kappa\lambda I_{\mathcal{H}}$.

We also use the auxiliary notation $G_t$ as in Appendix B of Pásztor et al. (2024), where

$$G_t(\boldsymbol{\theta}_1, \boldsymbol{\theta}_2) = \lambda I_{\mathcal{H}} + \sum_{i=1}^{t} \alpha(z_i; \boldsymbol{\theta}_1, \boldsymbol{\theta}_2)\phi(z_i)\phi^\top(z_i),$$

and

$$\alpha(z, \boldsymbol{\theta}_1, \boldsymbol{\theta}_2) = \int_0^1 \dot{\mu}\left(\nu\,\boldsymbol{\theta}_2^\top \phi(z) + (1 - \nu)\,\boldsymbol{\theta}_1^\top \phi(z)\right) d\nu$$

is the coefficient arising from the mean value theorem, such that

$$\mu(\boldsymbol{\theta}_2^\top \phi(z)) - \mu(\boldsymbol{\theta}_1^\top \phi(z)) = \alpha(z, \boldsymbol{\theta}_1, \boldsymbol{\theta}_2)(\boldsymbol{\theta}_2 - \boldsymbol{\theta}_1)^\top \phi(z).$$

See Pásztor et al. (2024, Lemma 11) for details. It then follows that

$$g_t(\boldsymbol{\theta}_2) - g_t(\boldsymbol{\theta}_1) = G_t(\boldsymbol{\theta}_1, \boldsymbol{\theta}_2)(\boldsymbol{\theta}_2 - \boldsymbol{\theta}_1), \tag{40}$$

as shown in the proof of Lemma 12 in Pásztor et al. (2024). We use this relation, along with the inequality

$$G_t(\boldsymbol{\theta}_1, \boldsymbol{\theta}_2) \succeq \kappa^{-1} V_t, \tag{41}$$

where $\succeq$ denotes the Loewner order, also from the proof of Lemma 12, in our analysis.

We use the notation $h(z) = \phi^\top(z)\boldsymbol{\theta}^\star$ for the underlying preference function and $\varepsilon_i = y_i - \mu(h(z_i))$ to represent the sequence of observation noise.

Inspired by the proof of confidence intervals in Vakili et al. (2021a), we express the prediction error as

$$
\begin{aligned}
|\mu(h_t(z)) - \mu(h(z))| &\le L\,|h_t(z) - h(z)| \\
&= L\left|\phi^\top(z)(\boldsymbol{\theta}_t - \boldsymbol{\theta}^\star)\right| \\
&= L\left|\phi^\top(z)\,G_t(\boldsymbol{\theta}^\star, \boldsymbol{\theta}_t)^{-1}\left(g_t(\boldsymbol{\theta}_t) - g_t(\boldsymbol{\theta}^\star)\right)\right| \\
&= L\left|\phi^\top(z)\,G_t(\boldsymbol{\theta}^\star, \boldsymbol{\theta}_t)^{-1}\left(\sum_{i=1}^t (y_i - \mu(h(z_i)))\,\phi(z_i) - \lambda\boldsymbol{\theta}^\star\right)\right| \\
&= L\left|\phi^\top(z)\,G_t(\boldsymbol{\theta}^\star, \boldsymbol{\theta}_t)^{-1}\left(\sum_{i=1}^t \varepsilon_i\,\phi(z_i) - \lambda\boldsymbol{\theta}^\star\right)\right| \\
&\le \underbrace{L\left|\phi^\top(z)\,G_t(\boldsymbol{\theta}^\star, \boldsymbol{\theta}_t)^{-1}\left(\sum_{i=1}^t \varepsilon_i\,\phi(z_i)\right)\right|}_{\text{Stochastic Term}} + \underbrace{L\lambda\left|\phi^\top(z)\,G_t(\boldsymbol{\theta}^\star, \boldsymbol{\theta}_t)^{-1}\boldsymbol{\theta}^\star\right|}_{\text{Bias Term}}
\end{aligned}
$$

The first line follows from the Lipschitz continuity of the sigmoid function. The second line uses the representer theorem to express $h_t(z) = \phi^\top(z)\boldsymbol{\theta}_t$ and $h(z) = \phi^\top(z)\boldsymbol{\theta}^\star$, where $\phi(z) = \Bbbk(z, \cdot)$, defined similarly to (Pásztor et al., 2024, Appendix A). The third line uses (40). The fourth line uses that $\boldsymbol{\theta}_t$ is the minimizer of the loss in Equation (36). The fifth line uses the notation $\varepsilon_i = y_i - \mu(h(z_i))$ for the observation noise. Finally, the expression is split into a stochastic term and a bias term, allowing us to follow the proof structure of the confidence bound in (Vakili et al., 2021a, Theorem 1).

**The stochastic term** is a sub-Gaussian random variable and can be bounded with high probability using standard concentration results. In particular, the sub-Gaussian parameter is determined by the norm of the coefficients applied to the independent noise terms $\varepsilon_i$, which are $1/2$-sub-Gaussian. This follows from the fact that $\varepsilon_i = y_i - \mu(h(z_i)) \in [-\mu(h(z_i)), 1 - \mu(h(z_i))]$, and therefore the noise sequence has bounded support of length 1.

$$
\begin{aligned}
\frac{1}{2}L\left\|\phi^\top(z)\,G_t(\boldsymbol{\theta}^\star, \boldsymbol{\theta}_t)^{-1}\boldsymbol{\Phi}_t\right\| &\le \frac{1}{2}L\|\phi(z)\|_{G_t(\boldsymbol{\theta}^\star, \boldsymbol{\theta}_t)^{-1}}\|\boldsymbol{\Phi}_t G_t(\boldsymbol{\theta}^\star, \boldsymbol{\theta}_t)^{-1}\boldsymbol{\Phi}_t^\top\|_{\mathrm{op}}^{1/2} \\
&\le \frac{1}{2}L\kappa\|\phi(z)\|_{V_t^{-1}}\|\boldsymbol{\Phi}_t V_t^{-1}\boldsymbol{\Phi}_t^\top\|_{\mathrm{op}}^{1/2} \\
&\le \frac{1}{2}L\sqrt{\frac{\kappa}{\lambda}}\sigma_t(z),
\end{aligned} \tag{42}
$$

where $\|\cdot\|_{\mathrm{op}}$ denotes the operator (spectral) norm. The first inequality follows from matrix arithmetic and the definition of operator norm. The second uses (41). The third uses the identity $\|\phi(z)\|_{V_t^{-1}} = \frac{1}{\sqrt{\lambda\kappa}}\sigma_t(z)$ (see, e.g., Pásztor et al., 2024), along with $\|\phi_t V_t^{-1}\phi_t^\top\|_{\mathrm{op}} \le 1$, which follows from the eigenvalue bounds of $\phi_t\phi_t^\top$ and $V_t^{-1}$.

Therefore, by the concentration inequality for sub-Gaussian random variables (see, e.g., Vershynin, 2018), with probability at least $1 - \delta$,

$$L \left| \boldsymbol{\phi}^\top(z) \, G_t(\boldsymbol{\theta}^\star, \boldsymbol{\theta}_t)^{-1} \left( \sum_{i=1}^{t} \varepsilon_i \, \boldsymbol{\phi}(z_i) \right) \right| \leq \frac{1}{2} L \sqrt{\frac{\kappa}{\lambda}} \sigma_t(z) \sqrt{2 \log(2/\delta)}.$$

**The bias term** is bounded as:

$$\begin{aligned} L\lambda \left| \boldsymbol{\phi}^\top(z) \, G_t(\boldsymbol{\theta}^\star, \boldsymbol{\theta}_t)^{-1} \boldsymbol{\theta}^\star \right| &\leq L\lambda \|\boldsymbol{\phi}(z)\|_{G_t(\boldsymbol{\theta}^\star, \boldsymbol{\theta}_t)^{-1}} \|\boldsymbol{\theta}^\star\|_{G_t(\boldsymbol{\theta}^\star, \boldsymbol{\theta}_t)^{-1}} \\ &\leq L\lambda\kappa \|\boldsymbol{\phi}(z)\|_{V_t^{-1}} \|\boldsymbol{\theta}^\star\|_{V_t^{-1}} \\ &\leq LB\sigma_t(z), \end{aligned} \tag{43}$$

where the second line uses (41), and the third line uses $\|\boldsymbol{\phi}(z)\|_{V_t^{-1}} = \frac{1}{\sqrt{\lambda\kappa}} \sigma_t(z)$, as discussed above. It also uses the bound $\|\boldsymbol{\theta}^\star\|_{V_t^{-1}} \leq \frac{1}{\sqrt{\lambda\kappa}} B$, which follows from:

$$\lambda \|\boldsymbol{\theta}^\star\|_{V_t^{-1}} \leq \frac{\lambda}{\sqrt{\lambda\kappa}} \|\boldsymbol{\theta}^\star\| \leq \sqrt{\frac{\lambda}{\kappa}} B, \tag{44}$$

where the first inequality follows from the fact that the smallest eigenvalue of $V_t$ is at least $\lambda\kappa$, and the second follows from the RKHS norm bound $\|\boldsymbol{\theta}^\star\| \leq B$.

Combining both bounds gives the following expression for $\beta(\delta)$:

$$\beta(\delta) = LB + \frac{L}{2} \sqrt{\frac{2\kappa}{\lambda} \log(2/\delta)}. \tag{45}$$

# D. Experimental Details

In this section, we provide details on the experimental setting. We describe the RKHS test functions, the Ackley function, and the Yelp Open Dataset used in our experiments. Additionally, we outline the selected hyperparameters and the computational resources utilized in our simulations. We also present the MaxMinLCB algorithm of Pásztor et al. (2024).

**RKHS test functions:**  In Section 5, we outlined the procedure for generating the test function $f$ as an arbitrary function within the RKHS of a given kernel. In Figure 3, we display the test functions generated in the RKHS for the SE kernel and the Matérn kernels with $\nu = 2.5$ and $\nu = 1.5$. The figure includes plots of the utility function $f$, the preference function $h(x, x') = f(x) - f(x')$, and the probability of preference $\mu(h(x, x'))$.

**Ackley test function:**  It is defined as follows (with $d = 1$ and $\mathcal{X} = [-5, 5]$):

$$f(x) = -20 \exp\left(-0.2 \sqrt{\frac{1}{d} \sum_{i=1}^{d} x_i^2}\right) \exp\left(\frac{1}{d} \sum_{i=1}^{d} \cos(2\pi x_i)\right) + 20 + \exp(1)$$

The preference function $h$ (difference in utilities) is then scaled to the range $[-3, 3]$. The Ackley function is shown in Figure 3.

**Yelp Dataset**  We use a subset of the Yelp Dataset, filtering it to include only restaurants in Philadelphia, USA, with at least 500 reviews and users who review at least 90 restaurants. The final dataset consists of 275 restaurants, 20 users, and 2563 reviews. Reviews for each restaurant are concatenated and processed using OpenAI's `TEXT-EMBEDDING-3-LARGE` model to generate 32-dimensional vector embeddings, which serve as the action set in the BOHF framework. User ratings (ranging from 1 to 5) are considered as the utility function $f$, which are then scaled to the range $[-3, 3]$. Missing ratings are handled using collaborative filtering. In each experimental run, we sample a random user from the set of 20 and conduct the experiment independently. We average the regret over 60 runs to produce the final plot.

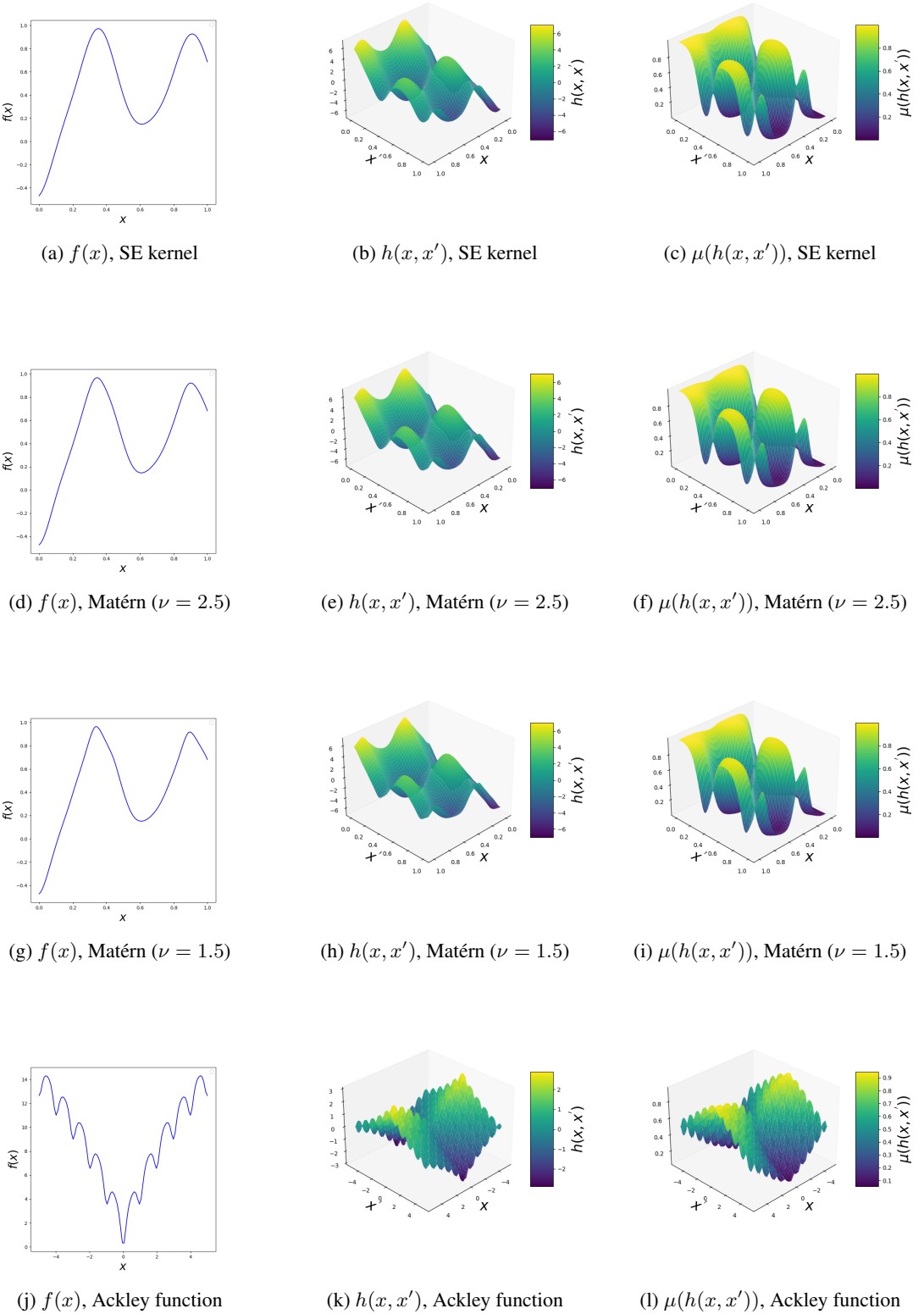

(a) $f(x)$, SE kernel

(b) $h(x, x')$, SE kernel

(c) $\mu(h(x, x'))$, SE kernel

(d) $f(x)$, Matérn ($\nu = 2.5$)

(e) $h(x, x')$, Matérn ($\nu = 2.5$)

(f) $\mu(h(x, x'))$, Matérn ($\nu = 2.5$)

(g) $f(x)$, Matérn ($\nu = 1.5$)

(h) $h(x, x')$, Matérn ($\nu = 1.5$)

(i) $\mu(h(x, x'))$, Matérn ($\nu = 1.5$)

(j) $f(x)$, Ackley function

(k) $h(x, x')$, Ackley function

(l) $\mu(h(x, x'))$, Ackley function

*Figure 3.* Plots of the utility function $f(x)$, the preference function $h(x, x') = f(x) - f(x')$, and the probability of preference $\mu(h(x, x'))$ for synthetic experiments. The rows correspond to: **(1st row)** SE kernel (RKHS), **(2nd row)** Matérn kernel with $\nu = 2.5$ (RKHS), **(3rd row)** Matérn kernel with $\nu = 1.5$ (RKHS), and **(4th row)** Ackley function.

**Loss function optimization:** To minimize the loss function given in (9) and obtain the parameters $\boldsymbol{\theta}$, any standard optimization algorithm can be used. In our experiments, we employ gradient descent. The learning rate is individually tuned for each algorithm, kernel, and test function by selecting the best-performing value from the grid $\{0.01, 0.005, 0.001, 0.0005, 0.0001\}$ in each scenario.

**Hyperparameters:** We choose $l = 0.1$ as the length scale and $\lambda = 0.05$ as the kernel-based learning parameter across all cases. The horizon $T$ is set to 300 for RKHS test functions and 2000 for the Ackley function and the Yelp Dataset. For the RKHS and Ackley functions, the confidence interval width $\beta$ is fixed at 1 for both MR-LPF and MaxMinLCB. For the Yelp dataset, we conduct a grid search to tune $\beta$ over $\{0.01, 0.1, 0.5, 1, 2\}$ for both MR-LPF and MaxMinLCB algorithms. We determine $\beta = 2$ as optimal for MaxMinLCB and $\beta = 0.1$ for MR-LPF.

**Computational Resources:** For the experiments with the synthetic RKHS and Ackley functions, we utilize the Scikit-Learn library (Pedregosa et al., 2011) for implementing Gaussian Process (GP) regression. The code is executed on a cluster with 376.2 GiB of RAM and an Intel(R) Xeon(R) Gold 5118 CPU running at 2.30 GHz. In the case of the Yelp Dataset experiments, we use the BoTorch library (Balandat et al., 2020) and its dependencies, including GPyTorch (Gardner et al., 2018), which offer efficient GP regression tools with GPU support. The simulations are carried out on a computing node equipped with an NVIDIA GeForce RTX 2080 Ti GPU featuring 11 GB of VRAM, an Intel(R) Xeon(R) Gold 5118 CPU running at 2.40 GHz with 24 cores, and 92 GB of RAM.

**MaxMinLCB algorithm:** Pásztor et al. (2024) proposed a zero-sum Stackelberg (Leader–Follower) game for action selection, where the leader $x_t$ maximizes the lower confidence bound (LCB), and the follower $x_t'$ minimizes it, according to the following:

$$x_t = \arg \max_{x \in \mathcal{M}_t} \mu(h_t(x, x'(x)) - \beta_t \sigma_t(x, x'(x)),$$
$$x'(x) = \arg \min_{x' \in \mathcal{M}_t} \mu(h_t(x, x')) - \beta_t \sigma_t(x, x').$$

