# OpenReview forum: "Bayesian Optimization from Human Feedback: Near-Optimal Regret Bounds"
_ICML.cc/2025/Conference — ICML 2025 poster_

### Official Review · Reviewer_FPrS · 2025-03-12

**Overall Recommendation:** 3

**Summary:**

This paper studies the kernel-based decision making problems under preferential feedbacks. The author proposed the phased-elimination style algorithm (which is refered to as MR-LPF in the paper) for this problem, which leads to $O(\sqrt{T \gamma_T})$ cumulative regret, while the existing algorithm suffers from additional polynomial dependence of $T$ and the complexity term arising from the non-linearity of the link function.  The author also conducted a numerical experiment in a real-world motivated problem, although the baseline method is limited to the existing UCB-based algorithm.

**Claims And Evidence:**

The author claimed contributions of this paper are well supported by theoretical perspective, and I could not find any issues as far as I checked.

**Essential References Not Discussed:**

To my knowledge, the existing results relating to this paper are well discussed in Section 1.1.

**Experimental Designs Or Analyses:**

We only check the description of experimental sections in the main paper. The synthetic experiments conducted in this paper are a standard setup for the kernel bandit or Bayesian optimization field, and we did not find any issue in the setup. One concern is that the average regrets at the initial step (t = 0) are different. To make the fair comparison, I believe that the initial points of the experiments should be the same in this setup.

**Methods And Evaluation Criteria:**

The author's proposed method can be interpreted as preferential variants of the existing phased-elimination or batched pure exploration-based approach. It makes sense to propose such a method to achieve better theoretical performance.

**Other Comments Or Suggestions:**

- Line 646: (x - x')^2 -> |x - x'|^2
- Eq. (18): the second x_t should be modified to x_t'
- The definition of R (e.g., Theorem 4.1, Line 782): log log (T) -> log_2 log_2 (T)
- Line 817: O(log^d T) -> O(log^{d+1} T) (see, Vakilli's paper).
- The arguments, why the maximum information gain of the kernel Eq. (4) increases at the same rate as the kernel $k$, should be added for the reader who is not familiar with this field. I did not find any arguments in existing papers, including the existing Pásztor's paper.

**Other Strengths And Weaknesses:**

My main concern about this paper's weakness is the novelty. The algorithm construction itself seems to be the naive extension of the standard phased elimination. As far as I see the proof, the difference between the existing phased elimination is as follows:

1. To keep $k_\tau$ small, the learner has to allocate enough samples for the initial phase (the proof from Line 690).
2. The non-adaptive confidence bound for preferential feedback (Theorem 4.7).

I did not check the details of the point 2 so far; however, this part seems non-trivial as far as my knowledge, and does not follow directly from existing literature's technique. Therefore, my current score is leaning toward acceptance.

**Questions For Authors:**

Please see "Theoretical Claims" section.

**Relation To Broader Scientific Literature:**

This paper's results show improved results of the existing theoretical guarantees of kernel-based dueling bandits.

**Theoretical Claims:**

I checked all the theoretical claims except for the proof of Theorem 4.7 (Appendix C). The proof is based on the standard techniques of GP-bandits, and I could not find any questionable point regarding the correctness of the theoretical claims. On the other hand, one of my concerns is whether the algorithm's near optimality, which the author appeals to in the paper, is rigorously verified. The existing Scarlett's lower bound is given for the Gaussian reward, and its connection with the preferential feedback model, whose feedbacks are given based on the comparison of chosen pairs of query points and the realization of Bernoulli distribution, is not straightforward for me.
In my intuition, the near-optimality of $O(\sqrt{\gamma_T T})$ in this paper's setup is true (may be rigorously proved by modifying the KL divergence of the Gaussian in Scarlett's proof); however, I believe more careful arguments should be added to the revision if the author claims the optimality. If not, the presentation (including title) of this paper should be modified by mildening the claim relating to near-optimality.

---

> ### Author Rebuttal · Authors · 2025-03-28
>
> Thank you for the positive feedback and careful review of the technical material, which is truly invaluable for us. Below, we address your comments and questions, which we hope will enhance your evaluation of the paper.
>
> > Interpretation of the lower bound of Scarlett et al., 2017 provided for standard BO with scalar feedback.
>
> We appreciate your raising this subtle point. Here we provide a technical clarification on our result and the mentioned lower bound.
>
> Consider a standard BO setting with noisy scalar observations $o = f(x) + \varepsilon$ (using the notation at the beginning of Section 2.3), where noise terms $\varepsilon$ are zero-mean, independent, and identically distributed across samples. Suppose at each step we select two points $x_t$ and $x'_t$, but instead of observing scalar feedback $o_t$ and $o'_t$ (with $o_t=f(x_t)+\varepsilon_t$ and $o'_t=f(x'_t)+\varepsilon'_t$), we receive preference feedback as a Bernoulli random variable $y_t = I(o_t > o'_t)$. Under the Bradley–Terry–Luce model, this preference feedback corresponds exactly to the case where the noise difference $\varepsilon'_t - \varepsilon_t$ follows a logistic distribution, whose CDF is the sigmoid function: $P(\varepsilon'_t - \varepsilon_t < f(x)-f(x')) = \mu(f(x)-f(x'))$.
> To have this satisfied, the individual noise terms $\varepsilon_t$ can be Gumbel-distributed.
>
> Thus, the lower bound on regret in our BOHF setting should be at least twice the lower bound for standard BO with scalar observations under Gumbel noise. In our revision, we will explicitly state this as a lemma and provide all the details.
>
> We acknowledge your remark that Scarlett et al. (2017) specifically use Gaussian noise in their lower bound example construction. However, it is standard practice in the BO literature to compare regret bounds under general sub-Gaussian noise (including bounded noise) with the lower bound result from Scarlett et al. (2017); see, for example, Salgia et al., 2021 and Li & Scarlett, 2022 among many others.
>
> Following your comment, we will explicitly state this result as a lemma in Section 4 and clarify the noise distribution assumption from Scarlett et al. (2017) in the revision. We are happy to discuss this further or answer any additional questions.
>
> > Novelty:
>
> We agree with the reviewer’s understanding of our theoretical results. Within the multi-round structure of our algorithm, we carefully characterize the impact of curvature and successfully remove the dependency on $\kappa$. Moreover, we leverage Theorem 4.7 to further improve regret with respect to $T$. These algorithmic and analytical contributions result in regret bounds of the same order as standard BO, providing a substantial improvement over existing methods, whose bounds can become vacuous for many kernels of theoretical and practical importance.
>
> We support these results through synthetic experiments on RKHS functions, which closely align with our theoretical assumptions and corroborate our analytical results. Additionally, we show the utility of our algorithm using a real-world Yelp dataset, highlighting that we have introduced a practical algorithm accompanied by strong theoretical guarantees.
>
> > Other Comments Or Suggestions:
>
> *The average regrets at the initial step (t = 0) are different:* This is due to arbitrary implementation choices which does not have a significant effect on eventual performance seen in the figures. We will fix this in the revision.
>
> *Why the maximum information gain of the kernel Eq. (4) increases at the same rate as the kernel?* In Proposition 4 of Pásztor et al. (2024), it is shown (see their Appendix C.1) that the eigenvalues of the dueling kernel $\mathbb{k}$ are exactly twice those of the original kernel $k$. Since the maximum information gain scales with the decay rate of the kernel eigenvalues (Vakili et al., 2021b, Corollary 1), both kernels exhibit the same scaling of the information gain. This property was also implicitly used in Pásztor et al. (2024). Following your comment, we will make this point explicit and clear.
>
> We fixed the typos mentioned by the reviewer. Thanks.
>
> ---
> We again thank you for your invaluable technical comments, which significantly enhance the presentation of the work. We hope addressing your comments improves your evaluation of the paper and remain available for further discussion.

---

> > ### Comment · Reviewer_FPrS · 2025-04-01
> >
> > Thank you for the author's clarification. I understand that the author considers the reduction to the lower bound under the Gumbel noise sequence. However, I disagree with the following author's claim:
> >
> > > However, it is standard practice in the BO literature to compare regret bounds under general sub-Gaussian noise (including bounded noise) with the lower bound result from Scarlett et al. (2017); see, for example, Salgia et al., 2021 and Li & Scarlett, 2022 among many others.
> >
> > The existing works (including Salgia et al, 2021 and Li & Scarlett, 2022) show the regret upper bound under sub-Gaussian noise. This is enough to show the near-optimality of the algorithm since the regret upper bound under sub-Gaussian noise random variables also holds under Gaussian noise random variables.
> > On the other hand, the author's argument only provides the fact that the lower bound of the preferential setting is the same rate as the standard problem lower bound under Gumbel noise. This does not formally imply the optimality of the algorithm since the lower bound is only provided under Gaussian noise, not the more general sub-Gaussian noise.
> > Therefore, I strongly recommend that the author modify the claim relating to near-optimality. I believe this modification will not be critical to this paper's results.

---

> > > ### Author Response · Authors · 2025-04-03
> > >
> > > We agree with the reviewer and will make this point clear in the revision. Specifically, as noted above, and following your comment, we will explicitly state the reduction to scalar feedback with Gumbel noise as a lemma in Section 4. We will also clarify that a direct comparison with the lower bound in Scarlett et al. (2017) is not formally adequate, since their construction assumes Gaussian noise, whereas our model relies on preference feedback that corresponds to Gumbel noise. We refer to the connection only as an informal indication of the quality of our results, not as a formal order optimality claim.
> > >
> > > Thank you again for your detailed and technical review. We greatly appreciate your engagement with the review process, which we believe significantly improved the paper.

---

### Official Review · Reviewer_6Kza · 2025-03-14

**Overall Recommendation:** 3

**Summary:**

This paper studies Bayesian optimization (BO) with preference feedback, in which every time a pair of inputs are selected and only a binary preference feedback is observed. The paper incorporates preference feedback into a multi-round structure inspired by previous works and prove that the resulting algorithm achieves $\sqrt{\Gamma(T)T}$ regret which is nearly optimal.

**Claims And Evidence:**

The paper claims that the proposed algorithm for BOHF can achieve a regret upper bound of $\sqrt{\Gamma(T)T}$, which is nearly optimal and matches the order of regret for scalar-valued feedback. These claims are supported by rigorous theoretical analysis, and the regret bound appears to be (nearly) tight.
However, the experimental results show that the proposed method performs better only for very large horizons, which weakens the claim of practical superiority over existing methods like MaxMinLCB.

**Essential References Not Discussed:**

The following is an importance missing reference: “Neural Dueling Bandits. 2024”. The paper uses neural networks to model the nonlinear function in dueling bandits and is hence highly related to the current paper.

**Experimental Designs Or Analyses:**

The experimental results have a notable weakness: the proposed algorithm only outperforms the existing method of MaxMinLCB when the horizon is very large. This limits its practical applicability in scenarios with smaller observation budgets. So, the experiments do not convincingly demonstrate practical superiority across a range of settings.

**Methods And Evaluation Criteria:**

The methods are generally appropriate for the problem studies, as the paper effectively adapts previous techniques for batched Bayesian optimization to the preference feedback setting. However, there may be a practical limitation: the proposed algorithm does not utilize preference observations when selecting the queries within a batch, which could potentially improve performance. The method is evaluated based on regret from both selected arms, which is standard in the literature of dueling bandits and hence acceptable.

**Other Comments Or Suggestions:**

The paper would benefit from a clearer explanation of the technical challenges involved in the proof, as this would help establish the novelty of the theoretical contributions. Additionally, incorporating preference observations into the query selection process could potentially improve the algorithm's performance and address its main practical limitation (i.e., it may help the proposed algorithm perform better than MaxMinLCB even with small horizons).

**Other Strengths And Weaknesses:**

Strengths:
- The proposed algorithm achieves a near-optimal regret order, which is, to the best of my knowledge, the tightest regret bound for BOHF problems.

Weaknesses:
- The technical novelty is a concern, as the analysis seems to combine existing techniques without clearly discussing the technical challenges involved.
- The proposed algorithm does not use the preference observations when selecting queries in a batch, which may limit its practical performance.
- Experiments show poor performance for small horizons, which is a critical limitation for practical applications.

**Questions For Authors:**

- Could **the constants** in the regret bound for BOHF be worse than those for scalar-valued feedback? Intuitively, weaker feedback should degrade performance. Clarifying this could provide a more complete picture of the theoretical guarantees.
- The proposed algorithm does not make use of the preference observations when selecting queries within a batch? In practice, incorporating these observations should lead to better performance. Could the algorithm be extended to account for them?
- What are the technical challenges in the proof that distinguish this work from prior analyses (e.g., Li & Scarlett, 2022; Pasztor et al., 2024; Xu et al., 2024)? A clear explanation of these challenges would help establish the paper's novelty.

**Relation To Broader Scientific Literature:**

The paper builds on prior works in batched Bayesian optimization and dueling bandits (or preferential BO), particularly combining techniques from works such as Li & Scarlett, 2022, Pasztor et al., 2024, and Xu et al., 2024. The paper indeed makes an important theoretical contribution to the fields of Bayesian optimization and dueling bandits.

**Theoretical Claims:**

The most important strength of the paper is the theoretical results, since it provides a regret bound with the order of $\sqrt{\Gamma(T)T}$, which is near-optimal for the problem setting. I also appreciate the insights given in the theoretical analysis section (especially the one at bottom of page 6), which provides an explanation as to why the algorithm can achieve such a small regret.
However, the technical novelty of the theoretical analysis is a concern. From a high level, the analysis techniques appear to a combination of those from existing works (Li & Scarlett, 2022; Pasztor et al., 2024; Xu et al., 2024). A clearer explanation of the technical challenges in the proof would enhance the contribution.

---

> ### Author Rebuttal · Authors · 2025-03-28
>
> We thank the reviewer for the detailed, comprehensive, and constructive feedback. We are glad that you found the theoretical results strong and the provided insights useful. Below, we address the questions, which we hope will help clarify and enhance your evaluation of the paper.
>
> > Could the constants in the regret bound for BOHF be worse than those for scalar-valued feedback?
>
> We naturally expect the constants to be worse, as the reviewer suggested, given that standard BO provides stronger feedback. While it is beyond the scope of this rebuttal to characterize the constants precisely—we note that even in standard BO, despite its extensive literature, there is no clear characterization of such constants—we are inspired by your comment to run experiments comparing empirical regret. Specifically, we will compare the regret, measured in terms of the underlying function values, for our BOHF algorithm and standard BO algorithms such as GP-UCB and BPE. Intuitively and as expected, we anticipate observing lower empirical regret for standard BO.
>
> > The proposed algorithm does not make use of the preference observations when selecting queries within a batch? In practice, incorporating these observations should lead to better performance. Could the algorithm be extended to account for them?
>
> The confidence intervals are explicitly utilized at the end of each round to eliminate actions that are unlikely to be optimal, making effective use of preference predictions. However, incorporating preference predictions within each round introduces intricate statistical dependencies that invalidate our current analysis framework, including the validity of confidence intervals derived at the end of such rounds. In short, employing preferences both during and at the end of each round cannot be handled with our current theoretical tools and would require further theoretical development.
>
> Empirically, the strong eventual performance of our algorithm relative to the state of the art demonstrates that our current approach—using preferences solely at the end of each round—is already effective in practice.
>
> > Technical challenges:
>
> Although inspired by existing methods, we introduce notable algorithmic and analytical novelties. In particular, the multi-round structure enables careful control of the dependency on $\kappa$, as detailed in our analysis and discussed at the bottom of page 6. Moreover, this structure allows us to leverage the new confidence interval established in Theorem 4.7, further improving the regret bounds with respect to $T$.
>
> Together, these improvements contribute to our near-optimal performance guarantees—a substantial improvement over the state of the art, which often results in possibly vacuous regret bounds in many cases of theoretical and practical interest such as Matérn and Neural Tangent kernels.
>
> > Missing reference: “Neural Dueling Bandits. 2024”.
>
> Thank you for mentioning this missing reference. We will include a review in the revision. They consider a wide neural network for target function prediction, in contrast to the kernel methods used in BOHF. However, their results are closely related, as a wide neural network can be approximated by the Neural Tangent Kernel as the layer widths grow. Beyond this modeling difference, their algorithm also differs from ours as well as from those of Pásztor et al., 2024 and Xu et al., 2024. Their method, referred to as Neural Dueling Bandit UCB (NDB-UCB), selects one action as the maximizer of the prediction and the other as the maximizer of the UCB. They also introduce a variant where the second action is selected via Thompson sampling instead of UCB. Their performance guarantees are expressed in terms of the effective dimension of the neural network model (see their equation (4)), which resembles the complexity term $\Gamma(T)$ in BOHF. They assume that the effective dimension grows slower than $O(\sqrt{T})$, which is a limitation, as otherwise their regret bounds become vacuous. Their regret bounds also scale with the curvature parameter $\kappa$.
>
> *Verma, Arun, et al. "Neural Dueling Bandits: Preference-Based Optimization with Human Feedback." The Thirteenth International Conference on Learning Representations, 2025.*
>
> ---
> Thank you again for your positive and constructive review. We are happy to answer any further questions during the rebuttal period.

---

### Official Review · Reviewer_yQVc · 2025-03-15

**Overall Recommendation:** 3

**Summary:**

This paper proposes a new algorithm, Multi-Round Learning from Preference-based Feedback (MR-LPF), for Bayesian Optimization from Human Feedback (BOHF). MR-LPF achieves a significantly improved regret bound of $\tilde{O}(\sqrt{Γ(T)T})$, matching the optimal regret bounds of conventional Bayesian optimization and eliminating the dependency on the curvature of the link function present in prior work. The algorithm operates in rounds, selecting pairs of actions with maximum uncertainty for preference feedback and iteratively refining a set of potentially optimal actions using confidence bounds.

**Claims And Evidence:**

The theoretical claims regarding the regret bound of the MR-LPF algorithm are supported by rigorous mathematical proofs presented in the appendices. These proofs build upon established techniques in Bayesian optimization and kernel methods, providing a clear and convincing argument for the improved theoretical performance. However, the empirical evidence presented to demonstrate the practical superiority of MR-LPF compared to existing methods is less convincing. While the figures show MR-LPF achieving lower regret than MaxMinLCB in the tested scenarios, the improvement is often modest, and the figures do not clearly illustrate the theoretical order-of-magnitude improvement suggested by the regret bounds. Further experiments with a wider range of benchmark functions and baselines, potentially including statistical significance testing, would be needed to fully substantiate the claim of superior practical performance. Specifically, the paper lacks the comparsion with POP-BO (Xu et al., 2024), making the experiments less convincing.

**Essential References Not Discussed:**

As far as i know, there is no more related paper needing to be cited in the paper.

**Experimental Designs Or Analyses:**

1. While the Yelp dataset provides a real-world application, the experimental setup is relatively limited. As your initial answer mentions, the paper itself highlights the increasing importance of preference learning due to LLMs. However, no experiments directly involve LLMs. A more compelling demonstration of practical utility would involve a task where preference feedback is elicited from human interactions with an LLM, for example, prompt optimization or text summarization comparison. This would more closely align the experiments with the motivating applications.
2. A crucial comparison is missing: POP-BO (Xu et al., 2024). POP-BO is a direct competitor in the BOHF literature, and omitting it significantly weakens the empirical evaluation. Without this comparison, it's difficult to definitively assess MR-LPF's relative performance within the current state-of-the-art.

**Methods And Evaluation Criteria:**

Yes, the proposed methods and evaluation criteria generally make sense for the problem at hand.

**Other Comments Or Suggestions:**

N.A.

**Other Strengths And Weaknesses:**

## Strengths:
- The paper's main strength is its theoretical contribution. It provides a new algorithm, MR-LPF, for BOHF and proves a significantly improved regret bound. This matches the order-optimal regret bounds of conventional Bayesian optimization and, crucially, eliminates the dependence on the curvature of the link function (κ) that plagued prior work. This is a substantial theoretical advancement in the field.
- The MR-LPF algorithm is well-motivated and clearly presented. The multi-round structure, with its variance-based action selection and confidence-bound-based pruning, is intuitive and logically sound. The algorithm description is easy to follow.
- The paper provides detailed proofs for its theoretical claims, building upon established techniques in Bayesian optimization and kernel methods. The appendices contain the necessary mathematical derivations.

## Weaknesses:

- The one of the most significant weakness is the omission of a comparison with POP-BO (Xu et al., 2024), a direct and highly relevant competitor in the BOHF literature. This makes it difficult to assess the relative performance of MR-LPF within the current state-of-the-art.
- While the experiments show MR-LPF performing better than MaxMinLCB (only after a large number of queries), the improvement is often not dramatic, and the figures don't clearly showcase the theoretical order-of-magnitude improvement.
- Despite motivating BOHF with applications like prompt optimization for LLMs, the paper does not include any experiments directly involving LLMs. This creates a disconnect between the stated motivation and the empirical evaluation.

**Questions For Authors:**

1. Why was POP-BO (Xu et al., 2024) not included as a baseline in the empirical evaluation?
2. Could you clarify the derivation of the inequality in Equation (42) within the proof of Lemma C.1?

**Relation To Broader Scientific Literature:**

The key contribution of this paper, a tighter regret bound for Bayesian Optimization from Human Feedback (BOHF), builds upon and significantly improves existing work in several areas. It relates to the broader literature on conventional Bayesian optimization (BO), where order-optimal regret bounds of $\tilde{O}(\sqrt{Γ(T)T})$ have been established for algorithms like GP-UCB and GP-TS (Srinivas et al., 2009; Chowdhury & Gopalan, 2017). The paper directly addresses the limitations of prior BOHF algorithms, specifically MaxMinLCB (Pásztor et al., 2024) and POP-BO (Xu et al., 2024), which had weaker regret bounds $\tilde{O}((Γ(T)κ^2\sqrt{T})$ and $\tilde{O}((Γ(T)T)^{\frac{3}{4}})$ respectively). The paper's achievement of a regret bound matching that of conventional BO, despite using the weaker preference-based feedback, is a notable advancement, demonstrating that the same sample complexities are achievable. The multi-round structure of the MR-LPF algorithm is inspired by, but distinct from, the Batch Pure Exploration (BPE) algorithm (Li & Scarlett, 2022) in conventional BO. It also connects to, but distinguishes itself from, dueling bandits and reinforcement learning from human feedback (RLHF) literature by focusing on kernel-based settings rather than tabular or linear settings.

**Theoretical Claims:**

I have not thoroughly checked every step of the proofs, which are complex and contained in the appendices.

---

> ### Author Rebuttal · Authors · 2025-03-28
>
> We thank the reviewer for the comprehensive review and positive feedback on our work. We are glad that you found our theoretical contributions substantial and the algorithm well-motivated and clearly presented. Below, we respond to your questions and comments, which we hope will further clarify and improve the paper.
>
> > Comparison with POP-BO (Xu et al., 2024)
> > Question 1: Why was POP-BO (Xu et al., 2024) not included as a baseline in the empirical evaluation?
>
>
> Theoretically, as discussed in detail in the introduction and summarized in Table 1, POP-BO provides the weakest regret bound among existing methods. In contrast, we achieve near-optimal regret bounds, explicitly showing that the number of preferential feedback samples required to identify near-optimal actions matches the order of scalar-valued feedback samples. Both MaxMinLCB and POP-BO exhibit vacuous regret bounds for kernels of practical and theoretical importance, such as Matérn and Neural Tangent kernels—with POP-BO being the weaker of the two.
>
> Empirically, we selected MaxMinLCB as the comparative baseline because Pásztor et al. (2024) have already demonstrated that MaxMinLCB outperforms POP-BO as well as several heuristic approaches. Thus, our choice ensures that our algorithm is compared against the strongest and most relevant available baseline. We appreciate the reviewer highlighting this point and will clarify this rationale in the revision.
>
>
> > While the experiments show MR-LPF performing better than MaxMinLCB, the improvement is often not dramatic.
>
> We fully agree with this observation. Indeed, the main contribution of our paper is analytical, and we establish substantial theoretical improvements over existing methods. The theoretical bounds we provide are *upper bounds* on performance. Based on the lower bounds for standard Bayesian optimization from Scarlett et al. (2017), our results are essentially unimprovable. It remains an open question whether the weaker bounds of Pásztor et al. (2024) and Xu et al. (2024) are artifacts of their proof techniques or reflect fundamental limitations of their algorithms—that is, whether the suboptimal upper bounds in those works truly capture the algorithms’ performance or whether tighter bounds could be derived for their algorithms.
> Nonetheless, our experiments show that the empirical performance of our algorithm is consistently strong across a range of synthetic RKHS examples that closely follow the theoretical assumptions, as well as on a real-world Yelp dataset.
>
> The goal of these experiments is to demonstrate that our contributions are not merely theoretical—they also yield a practical and robust algorithm with solid empirical performance, consistently outperforming the current state-of-the-art. That said, the main focus of our paper is to establish the core theoretical result: that the same number of preference samples are sufficient to identify near-optimal actions as in standard Bayesian optimization with scalar-valued feedback.
>
> > Despite motivating BOHF with applications like prompt optimization for LLMs, the paper does not include any experiments directly involving LLMs. This creates a disconnect between the stated motivation and the empirical evaluation.
>
> We agree with the reviewer that our experiments do not directly involve learning tasks using LLMs—we only use OpenAI's text embeddings model to generate vector embeddings for Yelp reviews. However, the BOHF framework is strongly motivated by an expanding body of work closely related to dueling bandits and reinforcement learning from human feedback (RLHF). This motivation is further supported by the growing number of recent papers addressing the same or similar problems.
>
> Our method can also be directly applied to prompt optimization for LLMs. However, such experiments require extensive setup and thorough implementations, which we view as an important and promising direction for a separate future work.
>
> > Question 2. Could you clarify the derivation of the inequality in Equation (42) within the proof of Lemma C.1?
>
> The inequality follows from the Sherman–Morrison formula and a rearrangement of terms. We will add the intermediate steps here and carefully review the proof to include clearer explanations at each step.
>
> ---
> Thank you again for your positive and constructive review, which helps clarify and improve the presentation of the paper.

---

### Official Review · Reviewer_ntoB · 2025-03-15

**Overall Recommendation:** 4

**Summary:**

This paper proposes a Bayesian optimisation method with only human preference-based feedback instead of classical scalar values. The order-optimal sample complexities of conventional BO are recovered. That means the number of preferential feedback samples is of the same order as the number of scalar feedback.

**Claims And Evidence:**

The claims are supported by thorough analysis and some empirical studies.

**Essential References Not Discussed:**

N/A

**Experimental Designs Or Analyses:**

The comparative method is only MaxMinLCB. It would be better to include more related methods as baselines.

For Ackley and Yelp data, do they satisfy the RKHS assumption?

**Methods And Evaluation Criteria:**

Yes, but the way to handle preferences is not clear. In Section 3, the main algorithm is introduced but how to train the utility function kernel ridge regression model is not clear. How is the preference data used for the model training?

The comparative method is only MaxMinLCB. It would be better to include more related methods as baselines.

**Other Comments Or Suggestions:**

If the theoretical analysis is the main contribution of this work and algorithm design is not, I would suggest moving the section to subsection 2.4? The readers may mainly focus on the Section 4.

**Other Strengths And Weaknesses:**

One concern is the novelty of the proposed algorithm in Section 3 where the preference-based function learning is mainly based on the existing approaches reviewed in Section 2.3.

The other concern is the experimental evaluation where only one comparative method is given.

**Questions For Authors:**

Is there any difference between the proposed algorithm with the ones from the literature?

**Relation To Broader Scientific Literature:**

The main contribution of this work is the derived tighter performance guarantees compared with related works.

**Theoretical Claims:**

No, I did not check the proof process in detail. The claimed results look reasonable to me.

---

> ### Author Rebuttal · Authors · 2025-03-28
>
> We thank the reviewer for the positive feedback and constructive comments. Below, we provide detailed responses and will incorporate these suggestions to further improve the presentation of the paper.
>
>
> > In Section 3, the main algorithm is introduced but how to train the utility function kernel ridge regression model is not clear. How is the preference data used for the model training?
>
> The preference function prediction is described in Section 2.3. Specifically, we express the preference function in terms of parameters $\boldsymbol{\theta}$ as shown in Equation (8), and obtain these parameters by minimizing the regularized negative log-likelihood loss defined in Equation (9). From a theoretical standpoint, any optimization algorithm can be used for this minimization. In our experiments, we employ gradient descent, and we clarify the choice of learning rate under *Experimental Details* in the appendix. Thank you for highlighting this point—we will ensure that this clarification is made explicit throughout the paper, including in Section 3 following the algorithm description.
>
> > For Ackley and Yelp data, do they satisfy the RKHS assumption?
>
> The Ackley function is infinitely differentiable. By the equivalence between the RKHS of Matérn kernels and Sobolev spaces, it belongs to the RKHS of all Matérn kernels with smoothness parameter $\nu \ge 1.5$, which includes the Squared Exponential kernel. However, its RKHS norm is difficult to characterize explicitly. For the Yelp dataset, we cannot assert that the utility function belongs to the RKHS.
>
> We interpret our experimental setup as follows: the RKHS experiments (first row of Figure 1) exactly match our theoretical assumptions and support the theory; the Ackley function provides a diverse optimization landscape while still fitting the theoretical framework; and the Yelp dataset offers a real-world test of practical utility.
>
> > One concern is the novelty of the proposed algorithm in Section 3 where the preference-based function learning is mainly based on the existing approaches reviewed in Section 2.3.
> > Is there any difference between the proposed algorithm with the ones from the literature?
>
> The key novelty and contribution of our work is the performance guarantees (regret bounds and sample complexities) we establish, which achieve near-optimality. We establish the key result that the number of preference samples required to identify near-optimal actions matches the order of scalar-valued feedback samples. This is in sharp contrast to prior works—existing bounds become vacuous in many cases of practical and theoretical interest, such as Matérn and Neural Tangent kernels.
>
> To achieve these results, we propose the MR-LPF algorithm. While MR-LPF and its analysis build on the well-established literature on kernel methods and Bayesian optimization, it introduces important algorithmic and analytical innovations. In particular, the multi-round structure enables precise control over the dependency on $\kappa$, as detailed in our analysis, and allows us to leverage the new confidence interval in Theorem 4.7 to obtain tighter bounds with respect to $T$.
>
> While inspired by Li & Scarlett (2022)’s work on standard BO, our algorithm and analysis differ due to the reduced preference feedback model, which introduces new challenges we address both algorithmically and analytically—including removing the dependency on the curvature of the nonlinear link function and confidence intervals for kernel methods from preference feedback.
>
>
>
> > The experimental evaluation where only one comparative method (MaxMinLCB) is given.
>
> We chose MaxMinLCB as the baseline because it is one of the two existing BOHF algorithms with theoretical guarantees (along with POP-BO), and Pásztor et al. (2024) show that it outperforms POP-BO and several heuristic methods. This makes it the strongest and most relevant baseline for comparison.
>
> Our goal is to demonstrate that our algorithm not only significantly improves theoretical guarantees but also performs well empirically.
>
> ---
> We thank the reviewer for the positive feedback and constructive comments. We believe that addressing these points will significantly enhance the clarity and quality of our paper.

---

### Decision · Program_Chairs · 2025-05-01

**Decision:**

Accept (poster)

**Comment:**

The reviewers are in agreement that this paper makes a valuable contribution to the area of kernel bandits with preferential feedback.  In particular, a near-optimal regret bound is given via a batched algorithm, which is in analogy with the regular feedback setting but by no means a trivial extension.  I pointed out some technical concerns but they were ultimately minor, and no major concerns remain.  Please carefully incorporate the reviewer comments (and my comments) into the final version.